# pp32 and APRIL are host cell-derived regulators of influenza virus RNA synthesis from cRNA

**Kenji Sugiyama, Atsushi Kawaguchi, Mitsuru Okuwaki, Kyosuke Nagata***

Department of Infection Biology, Faculty of Medicine, University of Tsukuba, Tsukuba, Japan

**Abstract** Replication of influenza viral genomic RNA (vRNA) is catalyzed by viral RNA-dependent RNA polymerase (vRdRP). Complementary RNA (cRNA) is first copied from vRNA, and progeny vRNAs are then amplified from the cRNA. Although vRdRP and viral RNA are minimal requirements, efficient cell-free replication could not be reproduced using only these viral factors. Using a biochemical complementation assay system, we found a novel activity in the nuclear extracts of uninfected cells, designated IREF-2, that allows robust unprimed vRNA synthesis from a cRNA template. IREF-2 was shown to consist of host-derived proteins, pp32 and APRIL. IREF-2 interacts with a free form of vRdRP and preferentially upregulates vRNA synthesis rather than cRNA synthesis. Knockdown experiments indicated that IREF-2 is involved in in vivo viral replication. On the basis of these results and those of previous studies, a plausible role(s) for IREF-2 during the initiation processes of vRNA replication is discussed.

*For correspondence: knagata@md.tsukuba.ac.jp

**Competing interests:** The author declares that no competing interests exist.

## Introduction

The influenza A virus genome is composed of eight single-stranded viral RNA (vRNA) segments of negative polarity that form viral ribonucleoprotein (vRNP) complexes with viral RNA-dependent RNA polymerases (vRdRPs) and nucleocapsid proteins (NPs). Both transcription and replication of vRNA are catalyzed by vRdRPs consisting of three viral proteins: PB1, PB2, and PA. Transcription of the influenza virus genome is initiated in a primer-dependent manner. The cap-1 structure of cellular mRNA was recognized by PB2 (*Guilligay et al., 2008*), and the capped RNA is cleaved 10–15 bases downstream of the 5'-terminus by the endonuclease activity of PA (*Dias et al., 2009*; *Yuan et al., 2009*). This cleaved, short RNA with a 5'-cap structure serves as a primer for the initiation of transcription. After elongation of the nascent RNA chain, transcription is terminated by a virus-specific polyadenylation mechanism (*Poon et al., 1999*): vRdRP reaches the poly(U) stretch of the vRNA template adjacent to its 5'-end and is thought to slip there repeatedly, leading to the addition of a poly (A) tail at the 3'-end of the nascent viral RNA transcript. Replication of the viral genomic RNA takes place in a primer-independent manner and proceeds in two steps: in the first step, vRdRP synthesizes full-length RNA copies of positive polarity, termed complementary RNA (cRNA); and in the second step, progeny vRNAs are amplified from the cRNA template. Both cRNA and vRNA products from each replication step contain a 5'-triphosphate end group (*Hay et al., 1982*; *Young and Content, 1971*), indicating that both replications are initiated de novo.

vRdRP and the viral RNA template are minimal and essential viral factors for viral RNA synthesis (*Deng et al., 2005*; *Lee et al., 2002*). NP is thought to play roles in efficient RNA elongation (*Honda et al., 1988*) and regulation of vRdRP activity for replication (*Beaton and Krug, 1986*; *Newcomb et al., 2009*; *Portela and Digard, 2002*; *Shapiro and Krug, 1988*). These observations, which were provided using mainly cell-free systems, concord with the fact that an in vivo viral mini-

**eLife digest** The influenza or "flu" virus infects millions of people each year, with young children and elderly individuals most vulnerable to infection. The influenza virus stores its genetic material in the form of segments of single-stranded viral RNA. After the virus infects a cell, it replicates this genetic material in a two-part process. First, an enzyme made by the virus – called RNA polymerase – uses the viral genomic RNA as a template to form a "complementary" RNA strand (called cRNA). This cRNA molecule is then itself used as a template to make more viral genomic RNA strands, which can go on to form new viruses.

Exactly how viral genomic RNA is made from cRNA is poorly understood, although previous research had suggested that this process may also involve proteins belonging to the invaded host cell. However, these host proteins had not been identified.

By mixing virus particles with extracts from uninfected human cells, Sugiyama et al. have now found that two host proteins called pp32 and APRIL help viral genomic RNA to form from a cRNA template. Both of these proteins directly interact with the viral RNA polymerase.

Sugiyama et al. then reduced the amounts of pp32 and APRIL in human cells that were infected with the influenza virus. Much less viral genomic RNA – and so fewer new virus particles – formed in these cells than in normal cells. Further work is now needed to understand how the pp32 and APRIL proteins interact with viral RNA polymerase. This could eventually lead to the development of new treatments for influenza.

genome replicon system can be reconstituted by transfection of expression plasmids for PB1, PB2, PA, and NP and of a plasmid for vRNA driven by cellular RNA polymerase I (*Pleschka et al., 1996*). These viral factors catalyze cell-free viral RNA synthesis, but at a limited level. Furthermore, cell-free viral replication activity using viral factors could be observed in the presence of a large amount of viral factors (*Vreede and Brownlee, 2007*). In early cell-free studies using nuclei or nuclear extracts (NEs) prepared from infected cells, a robust level of transcription took place and replication to some extent could be observed (*Beaton and Krug, 1984*; *Jackson et al., 1982*; *Nagata et al., 1989*). Dissection and reconstitution of a cell-free viral RNA synthesis system using vRNP complexes isolated from virions, an exogenously added model RNA template, and uninfected NEs indicated that host-derived factors could be involved in the process of viral RNA synthesis (*Shimizu et al., 1994*). Using this assay system, we identified two stimulatory host factors, designated as RNA polymerase activating factors (RAF) -I and -II, which were identified as Hsp90 and UAP56/BAT1, respectively (*Momose et al., 2001*; *2002*). Later, we found a factor, designated as influenza virus replication factor (IREF)-1, which upregulates unprimed cRNA synthesis from vRNA by the vRNP complex through promotion of the promoter clearance step and avoidance of premature abortive RNA synthesis. IREF-1 was shown to be identical to the minichromosome maintenance (MCM) complex (*Kawaguchi and Nagata, 2007*). Recently, several screening studies including high-throughput screenings have indicated that there may be more candidates for the host factors that affect viral RNA synthesis (*Brass et al., 2009*; *Hao et al., 2008*; *Karlas et al., 2010*; *Konig et al., 2010*; *Naito et al., 2007*; *Watanabe et al., 2014*). Among these potentially relevant candidates, some were further examined and characterized using in vivo and/or cell-free analyses.

Here, we focused our study on identification of a novel host factor(s) that facilitates the efficient second step of replication, that is, vRNA synthesis from the cRNA template. We found that a crude fraction of uninfected NE enables vRdRP to synthesize unprimed vRNA from the cRNA template effectively, and we thus designated the novel factor(s) responsible for this activity as IREF-2. Further fractionation and purification identified two host-derived factors, pp32 and APRIL. The target of these factors was shown to be a free form of the vRdRP trimeric complex. IREF-2 was found to preferentially upregulate vRNA synthesis rather than cRNA synthesis. Knockdown (KD) of IREF-2 using short interfering dsRNA resulted in decreased levels of viral RNA in the infected cells. However, the primary transcription from the incoming vRNP complexes was not affected. These results suggest that IREF-2 functions as a host factor for vRNA synthesis from cRNA in infected cells.

## Results

### Purification of IREF-2 from NEs of uninfected cells

To establish a cell-free viral RNA synthesis system that mimics the second replication step, that is, unprimed vRNA synthesis from a cRNA template, we used micrococcal nuclease-treated vRNP complexes (mnRNP) as an enzyme source and the 53 nucleotide (nt)-long RNA harboring both terminal sequences of cRNA derived from segment 8 (designed 'c53') as an exogenous cRNA model template (*Shimizu et al., 1994*). This enzyme source was established in an early study (*Seong and Brownlee, 1992*) and well characterized in subsequent studies (*Galarza et al., 1996*; *Seong et al., 1992*). These previous studies showed that mnRNP does not support unprimed vRNA synthesis from the cRNA template. NEs prepared from uninfected cells were fractionated through a phosphocellulose column by stepwise elution with increasing concentrations of KCl. P0.05 was a fraction unbound to the column, whereas P0.2, P0.5, and P1.0 were fractions eluted using 0.2 M, 0.5 M, and 1.0 M salt solutions, respectively. These fractions were examined individually in terms of their ability to promote cell-free vRNA synthesis reaction in the absence of any primer (*Figure 1A*, lanes 3–6). The 53 nt-long RNA product was formed only in the presence of the P0.05 fraction (lane 3). The RNA synthesis was dependent on the mnRNP, c53, and P0.05 fractions (*Figure 1A*, lanes 8–10). This novel activity present in the P0.05 fraction was designated as IREF-2. Next, we tried to purify and identify the factor responsible for the IREF-2 activity in the P0.05 fraction through fractionation using sequential column chromatography (*Figure 1B*). To determine fractions containing the IREF-2 activity at each column chromatography step, every fraction was individually assayed for RNA synthesis ability in the presence of mnRNP and the c53 model template, but in the absence of any added primer. On

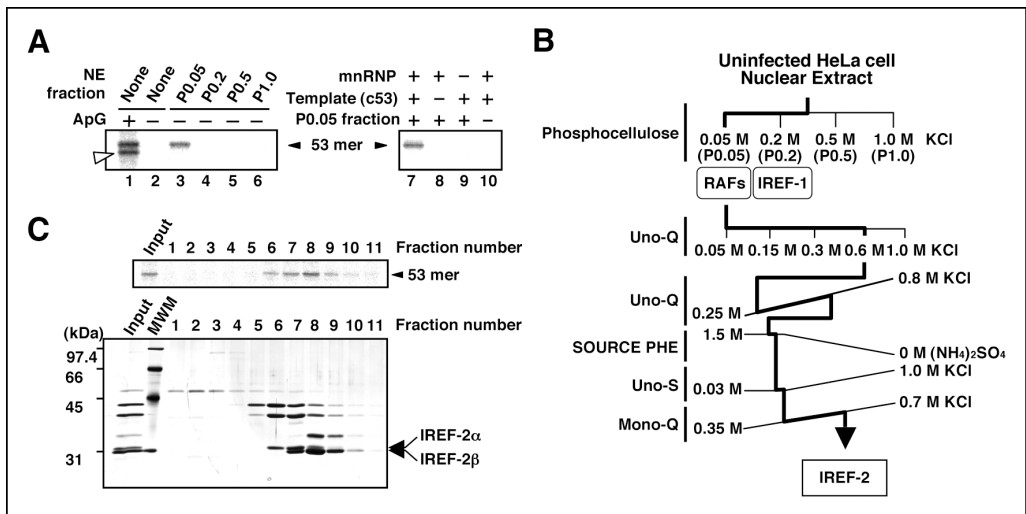

**Figure 1.** Purification of influenza virus replication factor-2 (IREF-2). (**A**) IREF-2 activity in uninfected nuclear extracts (NEs). Biochemical complementation assays using a cell-free vRNA replication system for fractions separated by phosphocellulose column chromatography were performed. The fractions, P0.05 (lane 3), P0.2 (lane 4), P0.5 (lane 5), and P1.0 (lane 6), were individually assayed in the cell-free viral RNA synthesis reaction employing 5 ng PB1-equivalent micrococcal nuclease-treated vRNP (mnRNP) as an enzyme source and 10 ng of the complementary RNA (cRNA) model template (c53), as described in the 'Materials and methods'. Dinucleotide ApG, serving as primer for viral RNA synthesis, was added to a final concentration of 0.2 mM (lane 1). To confirm the components required for the reactions, cell-free viral RNA synthesis with mnRNP and c53 in the presence of the P0.05 fraction was carried out (lane 7; identical to the conditions of lane 3). Simultaneously, reactions omitting the cRNA model template (lane 8), mnRNP (lane 9), or P0.05 fraction (lane 10; identical to the conditions of lane 2) were also carried out. After incubation at 30°C for 2 hr, each reaction product was collected and subjected to 10% Urea-PAGE followed by autoradiography. (**B**) Purification scheme of IREF-2 from uninfected HeLa cell NEs. For details regarding the column chromatography, see 'Materials and methods'. (**C**) Profile of the fractions from Mono-Q column chromatography at the final purification step. Each Mono-Q fraction (fraction numbers 1–11) or input material for the Mono-Q column chromatography (i.e., unbound fraction of the Uno-S column chromatography) was individually added to this cell-free viral RNA synthesis reaction in the absence of any added primers (*upper panel*). Each Mono-Q fraction was subjected to 11.5% SDS-PAGE, and polypeptides were visualized by silver staining (*lower panel*). The closed arrowhead indicates 53 mer RNA products. The open arrowhead (50 mer) indicates the product possibly generated by internal priming of ApG. The arrows indicate two candidate peptides responsible for IREF-2 activity. The molecular weight (kDa) positions are denoted on the left side of the panel.

the basis of its chromatographic behavior, IREF-2 appears to be highly acidic. At the final purification step using an anion exchanger (the Mono-Q column), the IREF-2 activity was recovered in fractions eluted with about 500 mM of KCl (fraction numbers 6–10 in *Figure 1C*, upper panel). The fractions from the Mono-Q column were also analyzed by SDS-PAGE followed by silver staining (*Figure 1C*, lower panel). Comparison of the polypeptide elution patterns and the level of the IREF-2 activity strongly suggested that two polypeptides with molecular masses of approximately 32 and 31 kDa corresponded to the IREF-2 activity (*Figure 1C*, arrows). The 32- and 31-kDa polypeptides were designated IREF-2$\alpha$ and $\beta$, respectively.

## Structure of IREF-2-dependent RNA products

To confirm the polarity of the IREF-2-dependent product, we carried out RNase T2 protection assays. ApG-primed RNA products using v53 and c53 as templates were also prepared as control materials (*Figure 2A*, lanes 1 and 4), in addition to the IREF-2-dependent RNA products (lane 7). Each radioactively labeled ApG-primed or unprimed 53 nt RNA product was hybridized with excess amounts of either the v53 or the c53 probe and then subjected to digestion with RNase T2, which preferentially digests ssRNA rather than dsRNA. Expectedly, the v53-directed ApG-primed RNA products were protected from digestion with RNase T2 by hybridization with the v53 probe (lane 2) but not the c53 probe (lane 3), whereas the c53-directed ApG-primed RNA products were protected by hybridization with the c53 probe (lane 6) but not with the v53 probe (lane 5), verifying the

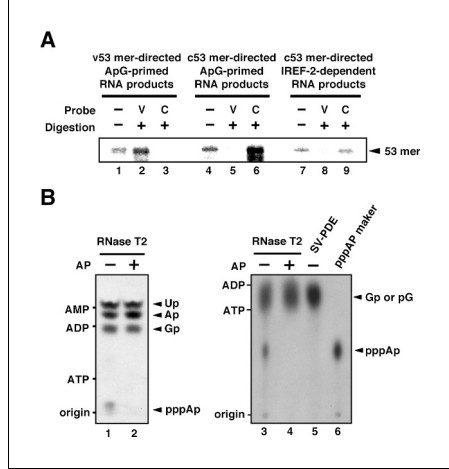

**Figure 2.** Products of influenza virus replication factor-2 (IREF-2)-dependent unprimed RNA synthesis. (**A**) RNase T2 protection assay. Radioactively labeled vRNA products were synthesized in the cell-free viral RNA synthesis system with micrococcal nuclease-treated vRNP (mnRNP) and the v53 model template in the presence of ApG (lanes 1–3), the c53 model template in the presence of ApG (lanes 4–6), and c53 in the presence of the IREF-2 fraction and in the absence of ApG (lanes 7–9). Viral RNA products were hybridized with excess amounts of nonlabeled v53 (lanes 2, 5, and 8) or c53 (lanes 3, 6, and 9), which was followed by digestion with RNase T2. Hybridized and digested RNA samples were extracted, collected, and subjected to 10% Urea-PAGE followed by autoradiography visualization. The closed arrowhead indicates 53-nt-long RNAs. (**B**) Analysis of the 5′-terminal structure of IREF-2-dependent unprimed vRNA products. [$\alpha$-$^{32}$P] GTP-labeled IREF-2-dependent unprimed vRNA products were prepared in the cell-free viral RNA synthesis system. The radioactively labeled 53-nt-long vRNA products were isolated from 10% Urea-PAGE, which was followed by excision and elution from the gel. A portion of the isolated 53-nt-long products was treated with alkaline phosphatase (lanes 2 and 4). Both nontreated and alkaline phosphatase-treated [$\alpha$-$^{32}$P] GTP-labeled unprimed vRNA products were digested with RNase T2 (lanes 1–4) or snake venom phosphodiesterase (lane 5). The digested materials were spotted onto a polyethylenimine (PEI)-cellulose thin layer and developed with 1 N acetic acid-4 M LiCl (4:1, v/v) (left panel; lanes 1 and 2) or 1.6 M LiCl (right panel; lanes 3–6) and visualized by autoradiography. For mobility standards, nonradiolabeled AMP, ADP, and ATP were also subjected to thin-layer chromatography and are indicated on the left side of each panel. For a marker of pppAp, [$\gamma$-$^{32}$P] ATP-labeled v53 synthesized using T7 RNA polymerase was also subjected to RNase T2 digestion and then to thin-layer chromatography (lane 6). The expected nucleotide positions are indicated on the right side of each panel by closed arrowheads.

authenticity of this assay. The IREF-2-dependent RNA product was protected by hybridization with the c53 probe (lane 9) but not with the v53 probe (lane 8). These results clearly indicate that the IREF-2-dependent RNA product is of negative polarity, that is, vRNA.

The other important point to be discussed regarding authentic vRNA replication is whether the vRNA product possesses a triphosphate moiety at its 5′-terminus due to de novo initiation. Thus, we analyzed the structure of the 5′-terminus of the IREF-2-dependent vRNA product (*Figure 2B*). The vRNA product synthesized in the presence of [α-$^{32}$P] GTP as a radioactive substrate in the reaction was expected to have 5′-xA[$^{32}$p]Gp. . .-3′. If the product is synthesized de novo, 'x' must be a triphosphate moiety (denoted as ppp). Upon digestion of the [α-$^{32}$P] GTP-labeled IREF-2-dependent vRNA product with RNase T2, which cleaves on the 3′ side of a phosphodiester bond, pppA[$^{32}$p] derived from the 5′-terminus could be detected by thin-layer chromatography separation (*Figure 2B*, lanes 1 and 3). This product, putative pppA[$^{32}$p], was not detected by treatment with alkaline phosphatase prior to RNase T2 digestion (lanes 2 and 4) or by digestion with snake venom phosphodiesterase (SV-PDE), which cleaves on the 5′ side of a phosphodiester bond (lane 5). These results clearly reveal that a triphosphate moiety is present at the 5′-terminus of the IREF-2-dependent vRNA product, thereby indicating that the synthesis of this vRNA product is initiated de novo. Taken together, these results indicate that the IREF-2-dependent unprimed vRNA product is a bona fide unprimed vRNA replication product from the cRNA template.

## Mass spectrometry analysis of IREF-2 proteins

Next, we attempted to reveal the identity of IREF-2α and β polypeptides. To this end, fraction numbers 6 (for IREF-2α, 32 kDa) and 9 (for IREF-2β, 31 kDa), shown in *Figure 1C*, were individually subjected to separation by SDS-PAGE, and polypeptides corresponding to IREF-2α (32 kDa) and β (31 kDa), were excised from the gel and then digested with trypsin. The trypsin-digested oligopeptides derived from IREF-2α and β were then analyzed using matrix-assisted laser desorption-ionization time-of-flight mass spectrometry (MALDI-TOF MS). By comparing the molecular masses of the oligopeptides from each IREF-2 protein obtained by MALDI-TOF MS analysis (see *Supplementary file 1*) with the database, we identified these IREF-2 proteins as follows: IREF-2α was shown to be identical to pp32 (phosphoprotein with a molecular mass of 32 kDa; accession number NP_006296) (*Malek et al., 1990*) and also known as leucine-rich acidic nuclear protein, inhibitor of protein phosphatase 2A (I1pp2A), and putative HLA class II-associated protein I (*Matilla and Radrizzani, 2005*). IREF-2β was shown to be identical to APRIL (acidic protein rich in leucines; accession number NP_006392) (*Mencinger et al., 1998*), also named proliferation-related acidic leucine-rich protein or silver-stainable protein 29 (*Matilla and Radrizzani, 2005*).

Both IREF-2α/pp32 and IREF-2β/APRIL exhibit high homology (71% sequence identity and 81% sequence similarity) and belong to the acidic nuclear phosphoprotein 32-kDa family (ANP32 family; pp32 and APRIL are also named ANP32A and ANP32B, respectively). These two proteins are encoded by separate genes and expressed mainly in the nuclei of variety kinds of tissues. Each IREF-2 protein is able to exist stably as a monomer and functions redundantly in multiple biological processes as an I1pp2A (*Li et al., 1996a*), a ligand to HuR that stabilizes ARE-containing mRNA (*Brennan et al., 2000*), a component for INHAT activity (*Seo et al., 2001*). The C terminus, comprising one-third of each protein, is extremely acidic, being composed of approximately 70% glutamic acid and aspartic acid residues, while the N-terminal region, comprising two-thirds of the protein, currently termed a leucine-rich-repeat, forms a horseshoe-shaped solenoid protein domain (*Huyton and Wolberger, 2007*; *Tochio et al., 2010*). By previous physical and regulatory mapping analyses, pp32 (ANP32A) was shown to be a host protein related to influenza virus infection (*Shapira et al., 2009*). In addition, a previous proteomic study identified pp32 and APRIL (ANP32B) as binding partners to the influenza vRdRP complex (*Bradel-Tretheway et al., 2011*). Recently, a genomewide RNAi screening study identified APRIL as one of nine 'top hit' genes affecting the viral RNA synthesis process in infected cells (*Watanabe et al., 2014*). However, functional analyses for IREF-2/ANP32s have not been carried out, and thus their functional importance remains unclear.

## Identification of pp32 and APRIL as IREF-2

Mono-Q chromatography could not separate IREF-2α/pp32 and IREF-2β/APRIL completely. Furthermore, several other polypeptides were also observed in these fractions (*Figure 1C*, fraction numbers

6–10). Therefore, we needed to confirm that pp32 and APRIL have authentic IREF-2 activity before any further molecular characterization. To this end, we prepared native and recombinant IREF-2 proteins and confirmed the quality of each protein preparation by silver staining and western blot analysis with anti-pp32 or anti-APRIL antibodies (*Figure 3—figure supplement 1*). Both native and recombinant IREF-2 proteins supported cell-free unprimed RNA synthesis using mnRNP and the c53 model template in a dose-dependent manner (*Figure 3A*, lanes 2–4 and 5–7, or lanes 8–11 and 12–15, respectively). These results clearly indicate that pp32 and APRIL are authentic proteins responsible for IREF-2 activity, able to function independently of each other. We also tested the IREF-2 activity of an acidic protein, TAF-I$\beta$/SET (*Nagata et al., 1995*). TAF-I$\beta$/SET was reported to exhibit similar functions to pp32 as an inhibitor of PP2A phosphatase (*Li et al., 1996b*) or INHAT (*Seo et al., 2001*). TAF-I$\beta$/SET did not exhibit any IREF-2 activity at all (lanes 16–18), suggesting that its acidic property is insufficient for IREF-2 activity.

We therefore next examined the template preference of IREF-2. Either v53 or c53 was used as the model RNA template for the cell-free reactions (*Figure 3B*). Notably, the unprimed RNA product

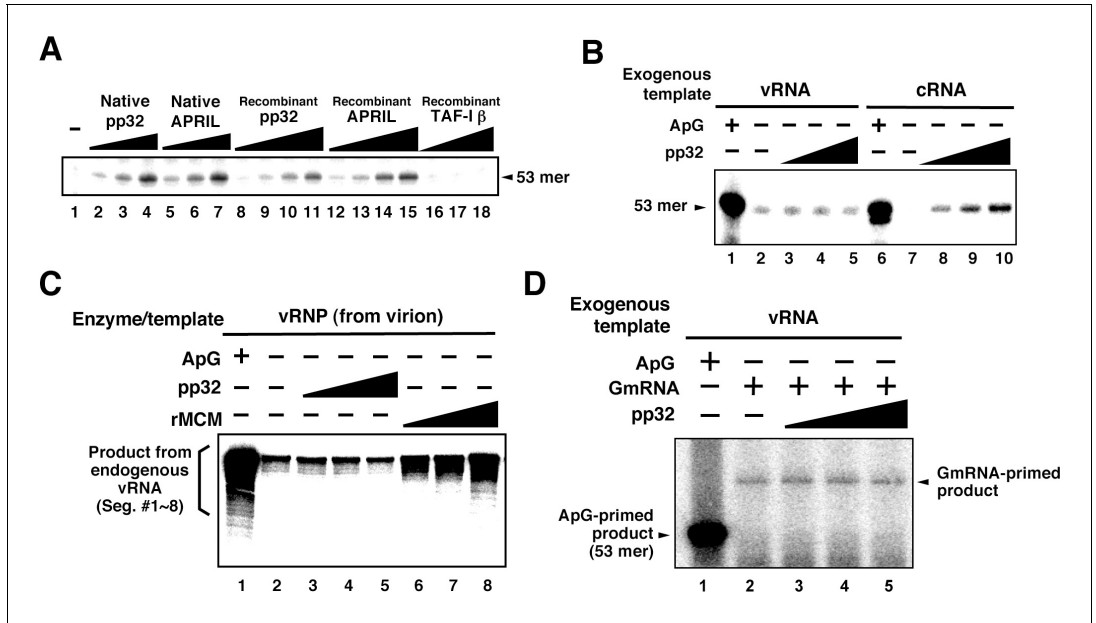

**Figure 3.** Characterization of influenza virus replication factor-2 (IREF-2) activities in the cell-free system. (**A**) Dose response of IREF-2. Native or recombinant IREF-2 proteins were added to the cell-free viral RNA (vRNA) synthesis reaction using micrococcal nuclease-treated vRNP (mnRNP) and the c53 model template in the absence of primer. Native pp32 (lanes 2–4), native APRIL (lanes 5–7), recombinant pp32 (lanes 8–11), and recombinant APRIL (lanes 12–15) were used as follows: 5 ng (lanes 2, 5, 8, and 12), 15 ng (lane 3, 6, 9, and 13), 50 ng (lanes 4, 7, 10, and 14) and 150 ng equivalent (lanes 11 and 15) of IREF-2 proteins. Recombinant TAF-I$\beta$/SET protein prepared using an *Escherichia. coli* expression system (33, 110, and 330 ng) was also tested (lanes 16–18). After incubation at 30°C for 2 hr, the RNA products were collected and analyzed by 10% Urea-PAGE followed by autoradiography. (**B**) Template preference of IREF-2-dependent viral RNA synthesis. Viral RNA replication reactions were performed in the cell-free viral RNA synthesis system using mnRNP and either v53 (lanes 1–5) or c53 (lanes 6–10) as viral model templates for vRNA and complementary RNA (cRNA), respectively. ApG at a final concentration of 0.2 mM (lanes 1 and 6) or 30, 100, and 300 ng of recombinant pp32 (lanes 3–5 and 8–10) was added to the reaction. (**C**) Effect of IREF-2 on cRNA synthesis from vRNP. Cell-free viral RNA synthesis using 2 ng PB1-equivalent of vRNP as the enzyme source and an endogenous genomic vRNA template were carried out in the presence (lane 1) or absence (lanes 2–8) of 0.2 mM ApG. Recombinant pp32 (lanes 3–5, 30, 100, and 300 ng, respectively) was added to the reactions. As a positive control, 1.5, 5, and 15 ng of recombinant IREF-1/MCM were also used (lanes 6–8). The RNA products were collected and analyzed by 4% Urea-PAGE followed by autoradiography. One-third (33%) of the total products derived from the ApG-primed cRNA synthesis were subjected to Urea-PAGE (lane 1). (**D**) Effect of IREF-2 on cap-snatching viral transcription. Cell-free viral RNA synthesis reactions were performed using mnRNP as the enzyme source and the exogenous model vRNA template (v53) in the presence of 0.2 mM ApG (lane 1) or globin mRNA as the 5′-capped RNA donor (lanes 2–5). Recombinant pp32 (lanes 3–5, 30, 100, and 300 ng, respectively) was added to the reaction.

The following figure supplement is available for figure 3:

**Figure supplement 1.** Protein profiles of native and recombinant influenza virus replication factor-2 (IREF-2s).

was observed from the cRNA template in an IREF-2-dependent manner (*Figure 3B*, lanes 8–10), but a significant level of cRNA synthesis was not detected when the vRNA template was used (lanes 3–5). These results suggest that IREF-2 preferentially regulates vRNA synthesis from the cRNA template, that is, the second step of the replication mechanism. Furthermore, the effect of IREF-2 on cRNA synthesis from the vRNP complex was also examined (*Figure 3C*). Cell-free viral RNA synthesis using vRNP complexes as the enzyme and endogenous vRNA template source, that is, the vRdRP and genomic vRNA of each segment, was performed in the absence of a primer. As previously observed, replicative cRNA synthesis from the genomic vRNA templates was stimulated by recombinant IREF-1/MCM (*Figure 3C*, lanes 6–8), which is known to stimulate promoter clearance during replication (*Kawaguchi and Nagata, 2007*). In contrast, no significant change induced by pp32 was observed (lanes 3–5). This finding indicates that IREF-2 is not involved in cRNA synthesis from endogenous vRNA and that the function of IREF-2 is distinct from that of IREF-1/MCM. This observation also confirms the template polarity preference of IREF-2 (*Figure 3B*).

Next, to address whether IREF-2 affects viral transcription, cap-snatching viral transcription was performed in the cell-free system using 5'-capped mRNA of β-globin (GmRNA) as a 5'-cap donor for viral transcription (*Figure 3D*). By adding GmRNA instead of ApG, the transcriptional product could be detected at a 10–12 nt longer size than that of the vRNA template (compare lanes 1 and 2), consistent with the findings of a previous report (*Plotch et al., 1979*). After addition of pp32 to this cell-free transcription reaction, no obvious change was observed (lanes 3–5), suggesting that IREF-2 has neither a positive nor a negative effect on viral mRNA transcription.

## Interaction between IREF-2 and viral RNA polymerases

To determine a regulatory target(s) of IREF-2, we performed interaction assays using IREF-2 and viral factors related to the viral RNA synthesis process. To date, no report has been published showing that pp32 and APRIL bind directly to RNA. In fact, the interaction between IREF-2 proteins and the model RNA template was not observed (*Figure 4—figure supplement 1*). Therefore, we performed GST pull-down assays using lysates prepared from mammalian cells (HEK293T) expressing GST-tagged IREF-2 (GST-IREF-2). Forty-eight hours after transfection with GST-derivative expression plasmids, the cells were infected with influenza virus at a multiplicity of infection (MOI) of 3 for 6 hr, and the viral factors bound to the GST-protein were then analyzed. Each subunit of the vRdRP complex (PB1, PB2, and PA) was co-precipitated with both GST-pp32 and GST-APRIL, but NP was not (*Figure 4A*, lanes 2 and 3). Instead, only NP was co-precipitated with GST-RAF2p48, which functions as a chaperone for NP (*Momose et al., 2001*) (lane 4). To further confirm this, lysates prepared from cells exogenously expressing polymerases (*Figure 4B*, lanes 1–4) or NP (lanes 5–8) and GST-derivative proteins were also examined. Exogenously expressed trimeric complexes of the viral polymerases were co-precipitated with both GST-pp32 and GST-APRIL (lanes 2 and 3), but NP was not detected at all (lanes 6 and 7). GST-RAF2p48 interacted with NP, but not with any polymerase subunits (lanes 4 and 8). Thus, we concluded that the binding target of IREF-2 is the trimeric complex of vRdRP, not NP. Moreover, the fact that the interaction between IREF-2 and NP could not be observed in the infected cells suggested that IREF-2 might interact with a free form of vRdRP, but not with vRdRP, in RNP complexes in infected cells. To confirm this, the amount of viral RNA in the complex of IREF-2 and vRdRP was quantitatively determined. Complexes of GST-pp32 with vRdRP were prepared from infected cells by GST pull-down, as shown in *Figure 4A* (lane 2). The RNA pulled down with the complexes was then extracted, and the amount of viral RNA (here, segment 5) was determined by reverse transcription-mediated quantitative PCR (RT-qPCR). We found that 1 pmol of vRdRP complexed with GST-pp32 contains <0.5 fmol of segment 5 (vRNA, cRNA, and viral mRNA) only, suggesting that IREF-2 tends to interact with a free form of vRdRP in infected cells.

We also tried to determine which subunit of vRdRP is the binding target of IREF-2. Cells expressing GST-tagged derivative proteins and vRdRP trimeric complexes or binary subcomplexes were prepared and subjected to GST pull-down assays (*Figure 4C*). Here, a Flag epitope tag was attached to the PB1 subunit, and formation of vRdRP subcomplexes, such as a trimeric complex (lanes 1–3) and a binary complex of PB1-PB2 (lanes 4–6) and of PB1-PA (lanes 7–9), was confirmed by immunoprecipitation with anti-Flag antibodies ('Flag IP' panels). Neither of the binary subcomplexes was pulled down with GST-IREF-2 ('GST pull-down' panels, lanes 5, 6, 8, and 9), whereas the interaction with IREF-2 could be observed for the vRdRP trimeric complex (lanes 2 and 3). A similar characteristic interaction property could be observed for each endogenous IREF-2 protein (*Figure 4—figure*

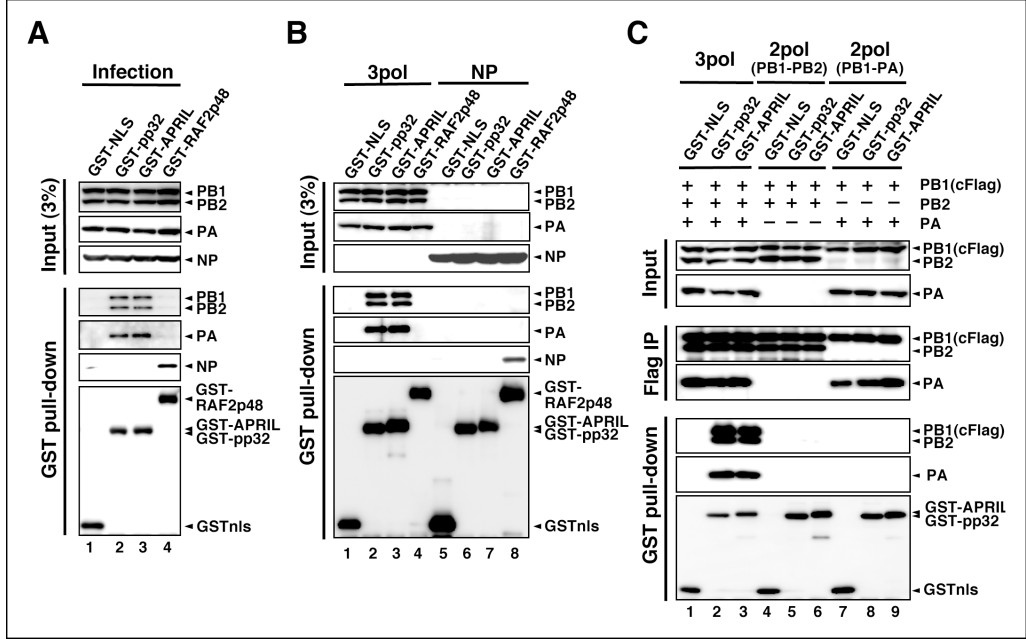

**Figure 4.** Interaction of influenza virus replication factor-2 (IREF-2) proteins with free forms of viral polymerase trimeric complexes. (**A**) Interaction between IREF-2 and viral proteins in infected cells. HEK293T cells were transfected with 10 μg of plasmids expressing GSTnls (lane 1), GST-pp32 (lane 2), GST-APRIL (lane 3), and GST-RAF2p48 (lane 4). At 48 hr post transfection, influenza virus was infected at an multiplicity of infection (MOI) of 3. At 6 hr post infection, the transfected and infected cells were collected, lysed, and subjected to GST pull-down assays, as described in 'Materials and methods'. The pulled-down materials and 3% equivalent of the input samples were subjected to SDS-PAGE followed by western blot analysis with anti-PB1, -PB2, -PA, -NP, and -GST (only the pull-down sample) antibodies. (**B**) Interaction between IREF-2 and viral proteins in the transfected cells. HEK293T cells were transfected with 5 μg of plasmids expressing GSTnls (lanes 1 and 5), GST-pp32 (lanes 2 and 6), GST-APRIL (lanes 3 and 7), and GST-RAF2p48 (lanes 4 and 8) and also co-transfected with the plasmids for the viral polymerase subunits (5 μg of pCAGGS-PB1, 12.5 μg of pCAGGS-PB2, and 2.5 μg of pCAGGS-PA [lanes 1–4] or 10 μg of pCAGGS-NP for NP expression [lanes 5–8]). Forty-eight hours post transfection, the cotransfected cells were collected, lysed, and subjected to GST pull-down assays. The input samples (3%) and pulled-down materials were subjected to SDS-PAGE followed by western blot analysis. (**C**) Interaction between IREF-2 and trimeric or binary complexes of vRdRP. HEK293T cells were cotransfected with 5 μg of plasmids expressing GSTnls (lanes 1, 4, and 7), GST-pp32 (lanes 2, 5, and 6), GST-APRIL (lanes 3, 6, and 9), 5 μg of pCAGGS-PB1cFlag (lanes 1–9), 12.5 μg of pCAGGS-PB2 (lanes 1–6), and 2.5 μg of pCAGGS-PA (lanes 1–3 and 7–9). Forty-eight hours post transfection, the lysates from the cotransfected cells were subjected to GST pull-down assays or immunoprecipitation assays with anti-Flag antibody. The precipitated materials were subjected to SDS-PAGE followed by western blot analysis.

The following figure supplements are available for figure 4:

**Figure supplement 1.** Electrophoresis mobility shift assay for influenza virus replication factor-2 (IREF-2) and viral RNA.

**Figure supplement 2.** Interaction between viral RNA-dependent RNA polymerase (vRdRP) complexes and endogenous influenza virus replication factor-2 (IREF-2).

*supplement 2*). Consistent with this observation, each IREF-2 protein was previously reported to be associated with the vRdRP trimeric complex, but not with the binary complex of PB1-PA (*Bradel-Tretheway et al., 2011*). And interestingly, the PB1-PB2 subcomplex was also found not to be associated with IREF-2, as shown here. This result strongly suggests that both the PA and the PB2 subunits in the trimeric complex of vRdRP are requisite for stable interaction with IREF-2.

## Effect of IREF-2 on viral RNA synthesis in infected cells

Using a cell-free viral RNA synthesis system, we demonstrated that IREF-2 enables vRdRP to replicate vRNA from a cRNA template preferentially. Next, to show the function of IREF-2 in vivo, the effect of IREF-2 KD on viral RNA synthesis in infected cells was examined. By transfection with short RNAi duplexes targeting pp32 and APRIL, the expression levels of both proteins were strongly

blocked (*Figure 5A*). Both the control KD and the IREF-2 KD cells were infected with influenza virus, and the intracellular viral RNA level was measured. Upon KD of pp32 or APRIL, the levels of vRNA, cRNA, and viral mRNA decreased to about 50% to –80% of the control level in each infection period (*Figure 5B*). By silencing both pp32 and APRIL ('double-KD' cells; see *Figure 5A*, lane 6), each viral RNA level was additively decreased to about 25% to – 50% of the control level (*Figure 5B*). These results suggest that IREF-2, pp32, and APRIL redundantly play a positive role in viral RNA synthesis in infected cells. Although IREF-2 was shown to preferentially regulate vRNA synthesis from a cRNA template in a cell-free system, the decrease patterns of the vRNA, cRNA, and viral mRNA syntheses were comparable in the IREF-2 KD cells.

We tried to address whether the effect of IREF-2 on vRNA and cRNA syntheses could be distinguished by using a mini-genome replicon system with either the vRNA reporter (vNS-Luc) or the cRNA reporter (cNS-Luc) as the source of the viral genome (*Figure 5—figure supplement 1*). In both templates, the levels of three kinds of viral RNAs were shown to be significantly decreased by IREF-2 KD as observed in the infected IREF-2 KD cells (*Figure 5B*). On the basis of the assumption that IREF-2 preferentially stimulates vRNA synthesis rather than cRNA synthesis, the vRNA level is expected to be more reduced by IREF-2 KD than the cRNA level in these experiments. In the case of vNS-Luc as a viral genomic source, the vRNA level appeared to be slightly more decreased by IREF-2 KD (12% of the control) when compared with the decrease in the cRNA level (23% of the control). Meanwhile, all viral reporter RNA levels were shown to be decreased almost comparably (46%-–48% reductions) when cNS-luc was used. From these results, the difference in the reduction level of vRNA and cRNA was insufficient to distinguish the effects of IREF-2 KD on vRNA and cRNA syntheses in these reporter experiments. This could be interpreted as follows: vRNA serves as a template for cRNA synthesis and mRNA transcription, and cRNA serves as a template for vRNA synthesis. Therefore, a defect in a certain specific step of viral RNA synthesis would influence the other steps and would result in declines in the accumulation levels of all viral RNA species, as observed in a previous study (*Maier et al., 2008*).

Next, we tried to address whether IREF-2 plays a role in transcription. To examine the effect of IREF-2 KD on primary transcription, we used a translation inhibitor, cycloheximide. In infected cells, cycloheximide prevents the synthesis of viral proteins required for the viral RNA replication process, resulting in the blockage of cRNA and vRNA syntheses. Therefore, only viral mRNA synthesis from incoming vRNP is detectable in the presence of cycloheximide. Control KD and pp32 and APRIL double-KD cells (identical to lanes 3 and 6 in *Figure 5A*) were infected at high multiplicity (MOI of 10) for 3 hr in the presence or absence of cycloheximide. In the presence of cycloheximide, the accumulation level of the primary transcription appeared to be almost the same for the control KD and IREF-2 double-KD cells, while in the absence of cycloheximide, a significant reduction was observed in the IREF-2 double-KD cells (*Figure 5C*). These results demonstrate that IREF-2 had no effect on the primary transcription, consistent with the result presented in *Figure 3D* showing that IREF-2 had no effect on the cap-snatching transcription in the cell-free reaction system. Thus, it is quite possible that IREF-2 stimulates viral RNA replication in vivo. Nevertheless the function of IREF-2 in cRNA synthesis and vRNA synthesis was not distinguishable in these experiments.

Finally, the effect of IREF-2 KD on virus growth was examined by using human alveolar basal epithelial cells (A549 cell line). Upon silencing both IREF-2 proteins (IREF-2 double-KD), significant decrease in progeny virus production (approximately, 90% of reduction at each time point) could be observed (*Figure 5D*). This seems to be attributable to the deficiency in the viral RNA replication process caused by IREF-2 KD.

## Discussion

In this study, we have identified IREF-2 consisting of pp32 and APRIL as host factors, which upregulate the second step of influenza viral RNA replication, that is, vRNA synthesis from a cRNA template. These host factors are well known as members of the ANP32 family and exhibit high homology with each other. Both are evolutionarily conserved in many organisms, including birds and swine, and are ubiquitously expressed in a variety of tissues, including lung tissue. Recently, an small-interfering RNA (siRNA) screening study suggested that APRIL is a positive regulator of viral RNA biogenesis (*Watanabe et al., 2014*). In that report, the interaction between host proteins and viral factors singly expressed in mammalian cells was also examined, but interaction between APRIL

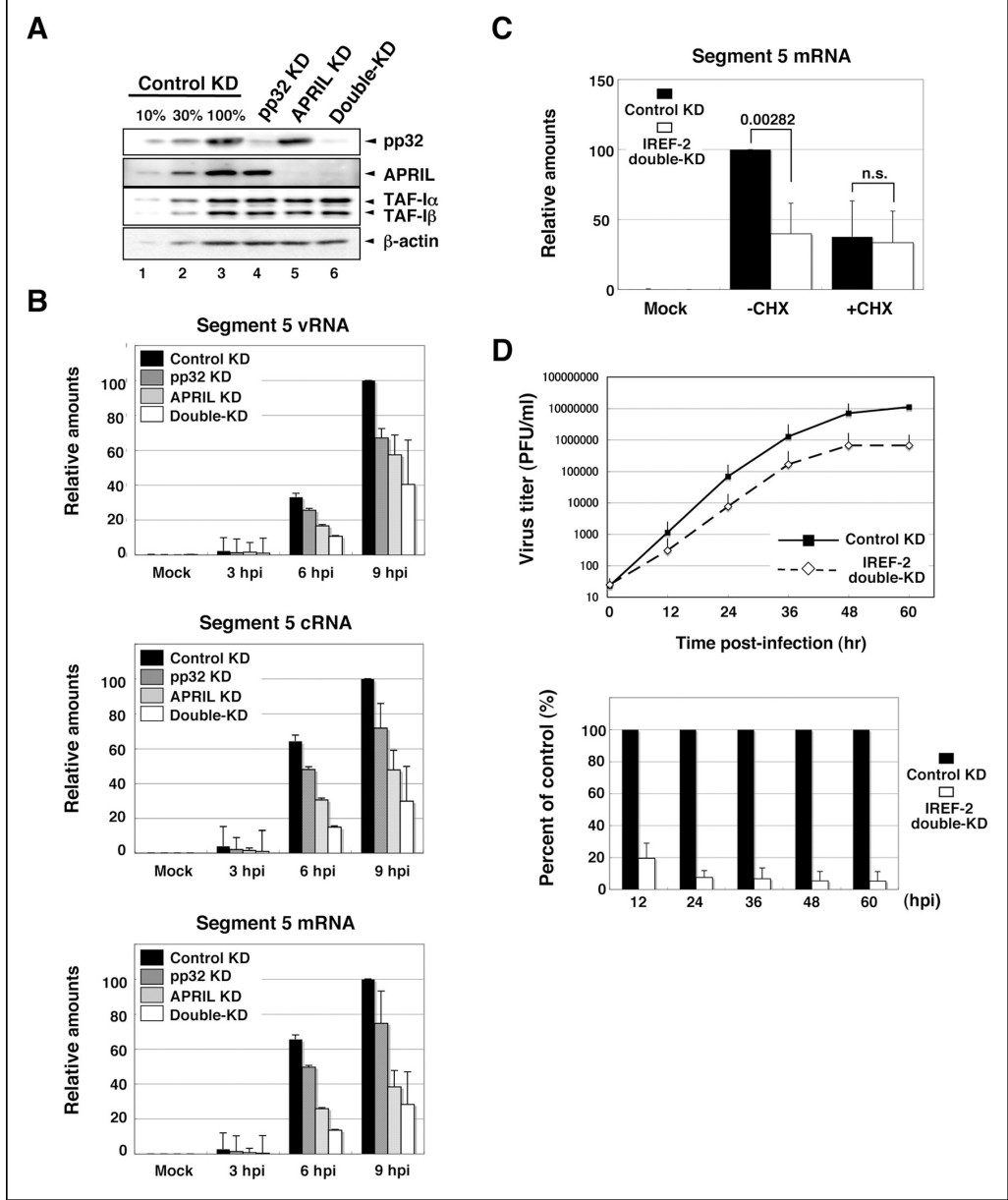

**Figure 5.** Effect of influenza virus replication factor-2 (IREF-2) knockdown in infected cells. (**A**) Endogenous IREF-2 protein levels in knockdown (KD) HeLa cells. The whole-cell lysates of control KD cells (lane 3), pp32 KD cells (lane 4), APRIL KD cells (lane 5), and both pp32 and APRIL KD cells (termed double-KD cells; lane 6) were subjected to SDS-PAGE followed by western blot analysis with antibodies, as indicated on the right side of the panel. To measure the KD levels, 10% and 30% of the lysates of the control KD cells were also subjected to SDS-PAGE followed by western blot analysis with antibodies, respectively (lanes 1 and 2). (**B**) Viral RNA levels in infected KD HeLa cells. Total RNA was extracted from mock-infected or infected KD cells at an multiplicity of infection (MOI) of 1 at 3, 6, and 9 hr post infection. vRNA, complementary RNA(cRNA), and viral mRNA derived from segment 5 were quantitatively determined by RT-mediated qPCR (RT-qPCR), as described in 'Materials and methods'. (**C**) Primary transcription level in infected KD HeLa cells. Total RNA was extracted from mock-infected or infected KD cells at an MOI of 10 for 3 hr in the presence or absence of cycloheximide (CHX). Viral mRNA derived from segment 5 was quantitatively determined by RT-qPCR. (**D**) Effect of IREF-2 on progeny virus production. Control KD and IREF-2 double-KD A549 cells were infected with FluV (WSN/33 strain) at an MOI of 0.001. The number of infectious progeny viruses produced from control KD and IREF-2 double-KD A549 cells at each time point were plotted (*upper panel*), and the ratios of progeny viruses produced from IREF-2 double-KD A549 cells compared with those from control KD cells were also represented as a percentage of the control (*lower panel*). Each quantitative result is presented as the average with the standard deviation from at least three independent experiments. Significance was determined using Student's *t* test. n.s.: not significant.

The following figure supplement is available for figure 5:

**Figure supplement 1.** Effect of influenza virus replication factor-2 (IREF-2) on viral reporter RNA syntheses from a reconstituted model replicon.

with vRdRP subunit was not observed, and the authors speculated that the effect of APRIL on viral RNA synthesis might be indirect. However, we have shown that IREF-2 can interact efficiently with the trimeric complex of PB1, PB2, and PA, but not with any subcomplexes (*Figure 4C*). Therefore, APRIL seems not to be co-precipitated with any singly expressed vRdRP subunit (*Watanabe et al., 2014*). It is noteworthy that interaction between IREF-2 and the vRdRP trimeric complexes was observed in infected cells and cells cotransfected with all three subunit expression vectors, while interaction with the RNP complexes could not be observed in the infected cells (*Figure 4A*). On the basis of these observations, the interaction between IREF-2 and vRdRP appears to be temporary during the process of cRNA template binding and vRNA synthesis, and once vRdRP starts vRNA synthesis, the associating IREF-2 protein may be released from vRdRP. The exact timing of the release of IREF-2 from the RNP complexes during replication, for example, at the initiation step of vRNA synthesis, during the transition step from initiation to elongation, or during elongation, remains to be solved. Cell-free analyses clearly demonstrated that IREF-2 is preferentially involved in vRNA synthesis from the cRNA template rather than in cRNA synthesis from the vRNA template (*Figures 3B, C*). The accumulation level of viral RNA in infected IREF-2 KD cells was significantly decreased, but not completely abolished (*Figure 5B*). This incomplete abolishment could be interpreted as due to the involvement of another unidentified host factor(s) in the viral RNA replication process. Recent studies have demonstrated that viral NS2 protein and virus-induced small leader RNA play important roles in the RNA replication process (*Perez et al., 2010*; *2012*; *Robb et al., 2009*). Taken together, these findings suggest that the negative effect of IREF-2 KD on viral RNA replication might be compensated by other host and/or viral factor(s) and/or other regulation mechanism(s) and that viral replication at some limited level could still be observed in IREF-2 KD cells.

Apparently, the function of IREF-2 is distinct from that of IREF-1/MCM (*Figure 3C*), which plays a role in promoter clearance during viral RNA synthesis, but how and in which step IREF-2 is involved is unclear. Previously, Deng et al demonstrated that the initiation steps of vRNA synthesis and cRNA synthesis take place in different ways: vRNA synthesis is initiated in an 'internal initiation and realignment' manner, whereas cRNA synthesis is initiated at the 3'-terminus of the vRNA template (*Deng et al., 2006b*). The reported initiation mechanism of vRNA synthesis in this model describes that dinucleotide pppApG is internally synthesized de novo so as to be complementary to $U^4C^5$ (in which the numbers indicate the nucleotide positions of the cRNA from the 3'-terminus and are denoted as 'c3'U4C5' in the discussion below). The 3'-terminal sequence of cRNA is composed of $3'-U^1C^2A^3U^4C^5\ldots$ -5'. Therefore, this internally synthesized pppApG would be realigned to the 3'-terminal nucleotide positions 1 and 2 (denoted as 'c3'U1C2') of the template cRNA, followed by subsequent extension resulting in the full-length vRNA product. On the basis of this model (*Deng et al., 2006b*), the differential effect of IREF-2 on vRNA and cRNA synthesis (*Figure 3B*) leads to the hypothetical speculation that IREF-2 functions at an initiation process specific to vRNA synthesis using cRNA as the template. In another previous report by Zhang et al, unprimed vRNA synthesis appears to be initiated de novo at the c3'U4C5 position of the cRNA template but 'realignment' does not take place. And vRdRP extends a nascent RNA chain from the internal position, resulting in a short vRNA product lacking the first three nucleotides at the 5'-terminal sequence of the authentic vRNA product (*Zhang et al., 2010a*). In their cell-free vRNA synthesis system, such abnormal 3 nt-shorter vRNA was a major product, whereas full-sized vRNA was synthesized only to a limited extent at high concentrations of the UTP substrate. In our study, however, IREF-2-dependent vRNA products were shown to be exclusively full length. Taking these results together, we postulated that IREF-2 regulates the initiation step of vRNA synthesis as follows: (1) by preventing improper extension prior to pppApG realignment, (2) by facilitating the conformational change of vRdRP and/or the cRNA template to promote the realignment, and/or (3) by stabilizing the initiation-intermediate complex when pppApG is already realigned onto c3'U1C2 and ready for proper elongation. To address these possibilities, further functional studies are needed.

In the two previous reports by Deng et al. and Zhang et al., cell-free vRNA synthesis could be reproduced without any exogenously added primer and host proteins, while in our cell-free system, no vRNA synthesis occurred without any added primer and host proteins, as was observed previously (*Galarza et al., 1996*; *Seong and Brownlee, 1992*; *Seong et al., 1992*). They used recombinant vRdRP purified from plasmid-transfected mammalian cells (*Deng et al., 2006b*) or baculovirus-infected insect cells (*Zhang et al., 2010a*; *2010b*), while we used mainly mnRNP prepared by treatment of virion-derived vRNP complexes with MNase as the enzyme source. Therefore, the intrinsic

nature of vRdRPs seems to differ one from the other, and recombinant vRdRP is possibly competent to initiate spontaneously de novo vRNA synthesis to some extent. Another difference among these enzyme sources is the presence of NP in the mnRNP preparation. At present, several lines of evidence suggest that NP plays a positive role(s) in regulation of vRdRP for replication and nascent chain elongation, but in the present study, unprimed vRNA synthesis employing mnRNP was not observed even in the presence of NP (lanes 2 and 10 in *Figure 1A*, and elsewhere). On the other hand, in previously reported preparations of recombinant vRdRP, some host proteins were possibly contaminated (*Deng et al., 2006a*; *Zhang et al., 2010b*), and an unexpected contribution(s) of such host protein contamination to their cell-free reactions could not be completely excluded. The other point for consideration is that the amount of vRdRP used in the cell-free reactions was different. In their cell-free reaction systems, a relatively large amount (100 nM) of vRdRP was used for the reaction (*Zhang et al., 2010a*), while we used a much smaller amount of the enzyme source (approximately 2–3 nM of vRdRP equivalent mnRNP) in the reaction. Hence, spontaneous vRNA synthesis in the absence of any primer and host factor might have taken place to some extent in their cell-free system. Taking such differences in the quality and quantity of the enzyme sources into account, it is possible that vRdRP intrinsically possesses de novo initiation activity and that IREF-2 may increase the turnover efficiency so that the apparent concentration of vRdRP seems to be increased. At this point, one of the primary interests to be addressed is whether IREF-2 exhibits similar stimulation activity as observed here in the cell-free system using recombinant vRdRP. In addition, the relationship between IREF-2 and NP has also not been addressed because mnRNP was used mainly as an enzyme source throughout this study. Therefore, NP-free vRdRP (i.e., recombinant vRdRP) seems to be a suitable enzyme source to address whether NP is necessary for the regulatory mechanism by IREF-2. These differences in the quality and quantity of vRdRP must be taken into account for further study. Lastly, we have shown by cell-free studies that IREF-2 is involved in regulation of unprimed vRNA synthesis but not of cRNA synthesis (*Figures 3B,C*). However, it is possible that our cell-free system might not be completely appropriate or sufficient for cRNA synthesis from a vRNA template, and thus we do not exclude the possibility that IREF-2 also functions in unprimed cRNA synthesis. If virion-derived vRdRP, including mnRNP, could be set up to be adapted to the transcriptional mode rather than to the (cRNA) replicative mode, some process might be required for neutralizing the transcriptional mode of vRdRP or switching the mode from 'transcriptase' to 'replicase'. Given this possibility, not only IREF-2 but also an additional factor(s) might be required for efficient unprimed cRNA synthesis in our cell-free system. To confirm this possibility, we are currently preparing recombinant vRdRP complexes possibly with the intrinsic property of influenza vRdRP as 'replicase' in infected cells for use in examining the effect of IREF-2.

In conclusion, we have demonstrated that the novel host-derived factors pp32 and APRIL regulate vRNA synthesis at least from the cRNA template. However, a number of questions about the function of IREF-2 remain unanswered. It would be important and beneficial to use recombinant vRdRP with high-purity and an authentic nature as an enzyme source in further detailed studies. In addition, the crystal structure of influenza vRdRP was recently resolved (*Pflug et al., 2014*; *Reich et al., 2014*), and therefore, the structure-based mechanism of IREF-2 will likely be elucidated in future studies.

## Materials and methods

### Cell culture, virus, and antibodies

Cells from the human cervical carcinoma HeLa S3 cell line were cultured in a spinner flask with Eagle's minimal essential medium for suspension cultures (S-MEM; Sigma, St. Louis, MO) containing 10% calf serum. The medium was added to the suspension cell culture to maintain a density of 2–$6 \times 10^5$ cells/ml. When the volume of the suspension cell culture reached 10 liters, the cells were collected by centrifugation and used for preparation of the NEs. HEK293T and A549 cells were grown at 37°C in Dulbecco's modified Eagle's medium (DMEM; Nissui, Japan) containing 10% fetal calf serum, and monolayer HeLa cells were cultured in Eagle's minimal essential medium (MEM; Sigma) containing 10% fetal calf serum, termed as MEM(+).

Influenza virus A/PR/8/34 (H1N1) was grown in the allantoic sacs of 10-day-old embryonated eggs. The viruses were purified from infected allantoic fluid as described previously (*Kawakami et al., 1981*) and used for vRNP preparation.

Anti-pp32 (goat polyclonal; Santa Cruz Biotechnology, Dallas, TX), anti-APRIL (goat polyclonal; Abcam limited, UK), and anti-GST (mouse polyclonal; Nacalai Tesque, Japan) antibodies were commercially purchased. For detection of viral proteins, rabbit polyclonal anti-PB1, -PB2, -PA, and -NP antisera were prepared as previously described (*Momose et al., 2001*; *2002*) and used for western blot analysis.

## Cell-free viral RNA synthesis

vRNP complexes were prepared from the purified influenza A/PR/8/34 virus as described previously (*Shimizu et al., 1994*). mnRNP was prepared by incubation of vRNP complexes at 30°C for 2 hr with one unit of micrococcal nuclease (Roche Molecular Biochemicals, Schweiz)/μl in the presence of 1 mM $CaCl_2$. The nuclease reaction was terminated by addition of EGTA to a final concentration of 3 mM. The 53 nt-long viral model RNAs (v53; 5'-AGUAGAAACAAGGGUGUUUUUUCAUAUCA UUUAAACUUCACCCUGCUUUUGCU-3' and c53; 5'-AGCAAAAGCAGGGUGAAGUUUAAAUGAUA UGAAAAAACACCCUUGUUUCUACU-3') were synthesized by transcription with RiboMAX Large Scale RNA Production System-T7 (Promega, Madison, WI) using a synthetic DNA template as previously described (*Shimizu et al., 1994*). Cell-free viral RNA synthesis was carried out at 30°C in a final volume of 20 μl or 25 μl in the presence of 50 mM HEPES-NaOH (pH 7.9), 3 mM $MgCl_2$, 50 mM KCl, 1 mM DTT, 500 μM each of ATP, UTP, and CTP and 25 μM GTP, 5 μCi of [α-$^{32}$P] GTP (3000 Ci/mmol), 8 U of RNase inhibitor from human placenta (Toyobo, Japan), 10 ng of a 53-nt-long model viral RNA template of negative or positive polarity (v53 and c53, respectively) and approximately 40 ng NP-equivalent, alternatively 5 ng PB1-equivalent of mnRNP as an enzyme source. After incubation, the reactions were terminated by phenol/chloroform extraction followed by precipitation of the RNA products with ethanol. The precipitated materials were subjected to polyacrylamide gel electrophoresis in the presence of urea (Urea-PAGE) and visualized by autoradiography.

## Purification of native IREF-2s from NEs

All procedures for preparation and fractionation of the NEs were carried out at 4°C or on ice. The details for purification is described at Bio-protocol (*Sugiyama and Nagata, 2016*). Briefly, uninfected HeLa cell NE was prepared as described previously (*Dignam et al., 1983*). For fractionation of the NE, a buffer (buffer H) containing 50 mM HEPES-NaOH (pH 7.9), 20% (v/v) glycerol, and 1 mM DTT in the presence of the appropriate concentrations of KCl was used for ion-exchange chromatography for purification of IREF-2. The purification scheme started with NE containing 15 mg protein. NE was loaded onto a phosphocellulose column (10-ml bed volume; Whatman P11) equilibrated with buffer H containing 50 mM KCl and successively eluted stepwise with buffer H containing 0.2 M, 0.5 M, and 1.0 M KCl. An unbound fraction (P0.05) and fractions eluted with 0.2 M (P0.2), 0.5 M (P0.5), and 1.0 M (P1.0) of KCl were examined by the cell-free viral RNA synthesis system. The remaining P0.05 fraction was loaded onto an anion exchanger Uno-Q column (Bio-Rad, Hercules, CA) equilibrated with buffer H containing 50 mM KCl and eluted stepwise with increasing concentrations of KCl. The IREF-2 activity fraction eluted with buffer H containing 0.6 M KCl was diluted with buffer H and again loaded onto the Uno-Q column equilibrated with buffer H containing 0.25 M KCl. The IREF-2 activity was eluted with a linear gradient of 0.25 M to –0.8 M KCl in buffer H. The concentrated IREF-2 fraction was diluted two-fold with an equal volume of a buffer containing 50 mM HEPES-NaOH (pH 7.9), 1 mM DTT, and 2.0 M $(NH_4)_2SO_4$ and loaded onto a hydrophobic SOURCE-PHE column (GE Healthcare, Piscataway, NJ) equilibrated with a buffer containing 50 mM HEPES-NaOH (pH 7.9), 10% (v/v) glycerol, 1 mM DTT, and 1.0 M $(NH_4)_2SO_4$. The IREF-2 activity was unbound to the hydrophobic column and dialyzed with buffer H containing 30 mM KCl. The dialyzed IREF-2 fraction was loaded onto a cation exchanger (Uno-S, Bio-Rad), and the IREF-2 activity was recovered in the fraction unbound to the column. Finally, the IREF-2 fraction was load onto an anion axcahnger (Mono-Q column; GE Healthcare), and the materials captured by the column were eluted with a linear gradient of 0.35–0.7 M KCl in buffer H.

## RNase T2 protection assay and thin-layer chromatography

For the RNase T2 protection assay, one-third of [α-$^{32}$P] GTP-radiolabeled total RNA products synthesized by the cell-free viral RNA synthesis reactions were hybridized with either a nonradiolabeled v53 or a c53 probe by incubation at 85°C for 10 min, followed by incubation at 60°C for 10 min in a hybridization buffer (40 mM PIPES-NaOH [pH 6.4], 1 mM EDTA, 0.4 M NaCl, and 80% formamide) and gradual cooling down to room temperature (Shimizu et al., 1994). Each hybridized RNA product was subjected to RNase T2 digestion (five units in 20 mM NaOAc [pH 5.2]) at 25°C for 1 hr. Digestion was terminated by phenol/chloroform extraction, and the digested materials were collected by ethanol precipitation. The precipitated materials were subjected to 10% Urea-PAGE and visualized by autoradiography. At the same time, the remaining one-third of the total RNA products was also subjected to Urea-PAGE as an undigested sample.

For analysis of the 5''-terminus of the unprimed IREF-2-dependent RNA products, RNase digestion and thin-layer chromatography were performed as described previously (Kawaguchi and Nagata, 2007). After purification by 10% Urea-PAGE and elution from the gel piece, the purified [α-$^{32}$P] GTP-radiolabeled RNA products were digested with 15 units of RNase T2 in 50 mM NaOAc (pH 5.0), 100 mM NaCl, and 2 mM EDTA, or 0.5 units of SV-PDE in 50 mM Tris-HCl (pH 9.0), 100 mM NaCl, and 14 mM MgCl$_2$ at 37°C for 1 hr. The digested products were spotted onto a PEI-cellulose thin layer (Merck, Germany) and developed with 1.6 M LiCl or 1 N acetic acid-4 M LiCl (4:1, v/v) and visualized by autoradiography. In the case of alkaline phosphatase treatment, [α-$^{32}$P] GTP-labeled product was incubated with 0.5 units of bacterial alkaline phosphatase in a buffer recommended by the manufacturer (Toyobo) at 37°C for 1 hr before nuclease digestion. For mobility standards, nonradioactive adenosine monophosphate (AMP), adenosine diphosphate (ADP), and adenosine triphosphate (ATP) were subjected to thin-layer chromatography. For a marker of adenosine 5'-triphosphate and 3'-monophosphate (pppAp), [γ-$^{32}$P] ATP-labeled v53 synthesized with T7 RNA polymerase was also subjected to gel purification, nuclease digestion, and thin-layer chromatography as described above.

## GST pull-down assay from mammalian cells

Transfected and infected HEK293T cells were harvested with a rubber policeman and washed with PBS(-). The collected cells were suspended in an ice-cold lysis-binding buffer (50 mM Hepes-NaOH [pH 7.9], 100 mM NaCl, 50 mM KCl, 0.25% NP-40, and 1 mM DTT) and lysed by brief sonication. After centrifugation, the crude lysates were incubated with Glutathione-Sepharose 4B resin (GE Healthcare) at 4°C for 1 hr. After incubation, the resins were collected by brief centrifugation and washed three times with the lysis-binding buffer. The resin-bound materials were eluted by boiling in the SDS-PAGE loading buffer and subjected to SDS-PAGE, which was followed by detection with the standard western blot analysis procedure.

## RNA isolation, reverse transcription, and quantification by RT- PCR

Total RNA from infected HeLa cells or transfected 293T cells was extracted using Sepasol-RNA I Super G (Nacalai Tesque) according to the manufacturer's instructions. The extracted RNA was treated with DNase I and subjected to RT reaction as follows. For synthesis of cDNA derived from segment 5 viral RNAs, each specific primer was used in the RT reaction, as follows: 5'-GACGATGCAACGGCTGGTCTG-3' (complementary to the 1122 nt-– 1142 nt region from the 5'-terminus) for the vRNA of segment 5; 5'-TCATCTTTGTTCCTCAA-3' (3' terminal region) for the cRNA of segment 5; and oligo-dT20 (5'-TTTTTTTTTTTTTTTTTTTT-3') for the viral mRNA. The RT reactions were performed using the hot-start protocol as described elsewhere (Kawakami et al., 2011). Total RNA (50–100 ng) and the aforementioned specific primer (10 pmol) were heated in a volume of 5.5 µl at 65°C for 5 min, chilled immediately in an ice-water bath, and then preincubated at 48°C. The RT premixture (3 µl First Strand buffer [5×, Invitrogen, Carlsbad, CA]; 0.75 µl of 0.1 M DTT; 0.75 µl of 10 mM dNTP mixture; 0.25 µl of Superscript III Reverse Transcriptase [200 U/µl, Invitrogen]; 0.25 µl RNase inhibitor; and 4.5 µl of saturated trehalose [Sigma] in a total of 9.5 µl) was also preincubated at 48°C and quickly added to the RNA-primer mixture and incubated at 48°C for 1 hr. The RT reaction mixture was diluted ten-fold with H$_2$O and incubated at 95°C to terminate the RT reaction. The diluted cDNA sample (3 µl) was subjected to quantification by real-time PCR using the FastStart SYBR Green Master (Roche) and the segment 5-specific primer set (5 pmol each/15 µl of the qPCR

reaction mix), 5′-GACGATGCAACGGCTGGTCTG-3′ and 5′-AGCATTGTTCCAACTCCTTT-3′. Real-time PCR was performed using the Thermal Cycler Dice Real Time System TP800 (Takara, Japan).

## Growth curve analysis

Growth curve analysis was performed by infecting confluent A549 cells (96 hr after transfection of siRNA) in 35-mm dishes with FluV/A/WSN/33 virus at an MOI of 0.001 and harvesting the supernatant every 12 hr after infection for 60 hr. The virus titers present in the supernatant were determined by a standard plaque assay using MDCK cells.

## Plasmid constructions

Full-length cDNA of human IREF-2α/pp32 was amplified using KOD polymerase (Toyobo) from a cDNA library prepared from HeLa cell mRNA using the specific primers 5′-CGCGGATCCCATA TGGAGATGGGCAGACGGATTCATTTAG-3′ and 5′-GCGGCTCGAGACGTCAGTCATCATCTTC TCCCTCATCTTCAGGTTCTCGT-3′, corresponding to the pp32 amino-terminal and carboxyl-terminal regions, respectively. The full-length cDNA of human IREF-2$\beta$/APRIL was also amplified by using the specific primers 5′-CGGAATTCATATGGACATGAAGAGGAGGATCCACCTGGAG-3′ and 5′-GCGGCTCGAGACGTCAATCATCTTCTCCTTCATCATCTGT-3′, corresponding to the APRIL amino-terminal and carboxyl-terminal regions, respectively. To construct the glutathione *S*-transferase (GST)-tagged pp32 expression vectors for *Escherichia. coli* cells (pGEX-2T-pp32), the amplified cDNA fragment of pp32 was digested with *Bam*HI and *Aat*II and then cloned into pGEX-2T (GE Healthcare) predigested with the same restriction enzymes. To construct *E. coli* expression vectors for GST-tagged APRIL (pGEX-2T-APRIL), the amplified cDNA fragment of APRIL was digested with *Eco*RI and *Aat*II and then cloned into pGEX-2T predigested with the same restriction enzymes. To construct mammalian expression vectors expressing GST harboring the nuclear localization signal (NLS) at the C-terminal end, termed pCAGGS-GSTnls, DNA fragments were amplified by PCR using the specific primer set 5′-CCCTCGAGCTCGCGGCCGCCGCCATGGGCTCCCCTATACTAGGTTA TTGG-3′ (the sequence responsible for the 'Kozak translational consensus' is underlined) and 5′-AAGATCTATGCATGGTACCGCTAGCGACGTCACCCGGGGTCTTCTACC-3′, and pGEX-2T-SV40 NLS-GFP (kindly gifted by Dr Yoneda, Osaka University) as the PCR template. The PCR products corresponding to Kozak-GSTnls were trimmed by digestion with *Xho*I and *Bgl*II and then cloned into the predigested pCAGGS mammalian expression vector, resulting in pCAGGS-GSTnls. To construct mammalian expression vectors expressing GST-tagged IREF-2s, pGEX-2T-pp32 and pGEX-2T-APRIL were digested with *Swa*I (for cutting within the GST coding region) and *Aat*II (for cutting downstream of the termination codon of both IREF2s), and each DNA fragment corresponding to GST$_{C-term}$-IREF-2 was subcloned into the pCAGGS-GSTnls predigested with *Swa*I and *Aat*II, resulting in pCAGST-pp32 and pCAGST-APRIL. To construct pCAGST-RAF2p48 expressing GST-fused RAF2p48/BAT1/UAP56, a DNA fragment corresponding to the GST-RAF2p48-encoding region derived from pGEX-p48 was subcloned into the pCAGGS vector, resulting in pCAGST-RAF2p48. The nucleotide sequences of all plasmids constructed for this study were confirmed by DNA sequencing (ABI prism genetic analyzer; Applied Biosystems, Foster City, CA).

## Preparation of recombinant IREF-2 proteins

To express recombinant IREF-2 proteins, *E. coli* BL21(DE3) cells harboring the expression vectors (pGEX-2T-pp32 and pGEX-2T-APRIL) for GST-tagged IREF-2s were cultivated at 30℃ until A590 reached 0.5, and protein production was then induced with 1 mM isopropyl-1-thio-$\beta$-D-galactopyranoside for 6 hr. The induced cells were harvested by centrifugation and resuspended in a lysis buffer (50 mM Hepes-NaOH [pH 7.9], 150 mM KCl, 0.25% NP-40, and 1 mM DTT) and sonicated well in an ice-water bath. The insoluble material was removed by centrifugation, and crude extracts were recovered as a supernatant fraction was applied to Glutathione-Sepharose 4B resin that had been equilibrated with the lysis buffer. After capturing of the GST-tagged proteins to the resin, lysis buffer was applied to wash away the unbound materials. The washed resins were equilibrated with a digestion buffer (50 mM Hepes-NaOH [pH 7.9], 50 mM KCl, 1 mM CaCl$_2$, and 1 mM DTT) and digested with thrombin (Nacalai Tesque) at 4℃ overnight. The materials with the GST portion removed were collected and further purified through a Mono-Q column with KCl linear gradient elution. The purified proteins were dialyzed with buffer H containing 1 mM DTT, and the concentration of each

protein was determined by the Bradford method and comparison of band intensities with a standard protein (BSA) after separation by SDS-PAGE.

## Transfection of DNA and siRNA and virus infection

For transfection of HEK293T cells with mammalian expression plasmids, plasmid DNA mixtures were prepared in Opti-MEM (Thermo Fisher Scientific, Waltham, MA), and an appropriate amount of PEI (Sigma Aldrich, St. Louis, MO) were mixed into the DNA solution (1.5 µg PEI per 1 µg DNA) and incubated at room temperature for 10 min. The DNA/PEI complex was added to monolayer cell cultures (approximately, $3 \times 10^6$ cells in a 100-mm-diameter dish). Six hours after transfection, the medium was replaced with fresh DMEM and maintained at 37°C.

For transfection of siRNAs, trypsinized cells (HeLa, HEK293T, and A549) were seeded, and pp32 siRNA (custom-designed Stealth RNAi; Invitrogen), APRIL siRNA (ANP32B-HSS116202; Invitrogen), and negative control siRNA (12935–200; Invitrogen) were introduced into the cells with Lipofectamine RNAiMAX (Invitrogen) according to the manufacturer's protocol. Twelve to twenty-four hours after the first transfection, the cells were again transfected with the same amount of siRNA used at the first transfection and maintained at 37°C for 4 to 5 days until influenza virus infection.

For virus infection, monolayer cultures of human cells (HEK293T, HeLa, and A549) were washed with serum-free MEM, and the cells were infected with influenza A/PR/8 virus allantoic fluid at the desired MOI as described in each figure legend. After virus adsorption at 37°C for 1 hr, the infecting allantoic fluid was removed, and the cells were washed with serum-containing medium and maintained at 37°C in the medium for an appropriate period (3–9 hr).

## Acknowledgements

The authors thank Dr Ryoich Kiyama (National Institute of Advanced Industrial Science and Technology, Tsukuba) for MALDI-TOFF mass analysis; Dr Yoshihiro Yoneda (Osaka University) for pGEX-2T-SV40 NLS-GFP; and Ms Flaminia Miyamasu (University of Tsukuba) for editing of the manuscript. This work was supported in part by grants-in-aid from the Ministry of Education, Culture, Sports, Science, and Technology of Japan (to KN and AK).

## Additional information

### Funding

| Funder | Grant reference number | Author |
|---|---|---|
| Ministry of Education, Culture, Sports, Science, and Technology | 24115002 | Kyosuke Nagata |

The funders had no role in study design, data collection and interpretation, or the decision to submit the work for publication.

### Author contributions

KS, Conception and design, Acquisition of data, Analysis and interpretation of data, Drafting or revising the article; AK, Acquisition of data, Analysis and interpretation of data; MO, Conception and design, Analysis and interpretation of data; KN, Conception and design, Analysis and interpretation of data, Drafting or revising the article

### Author ORCIDs

Kyosuke Nagata, http://orcid.org/0000-0003-2522-3561

## Additional files

### Supplementary files

• Supplementary file 1. Molecular masses of trypsin-digested peptides from IREF-2 proteins and amino acid sequences of IREF-2 proteins corresponding to the tryptic cleavage molecular mass database.

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
