## [Decision Letter]

Thank you for submitting your work entitled "pp32 and APRIL are host cell-derived regulators for the influenza virus vRNA synthesis from cRNA" for peer review at *eLife*. Your submission has been favorably evaluated by Detlef Weigel (Senior Editor), a Reviewing Editor, and three reviewers, one of whom is a member of our Board of Reviewing Editors. One of the three reviewers, Andrew Mehle (Reviewer #2), has agreed to reveal his identity.

The reviewers have discussed the reviews with one another and the Reviewing Editor has drafted this decision to help you prepare a revised submission.

Essential revisions:

While all the reviewers were excited by the findings, there were several perceived weaknesses in the story, and additional work was universally asked for. What was most critically needed was the inclusion of better indications of the importance of these factors in cells. Very specific experiments were proposed in the reviews, and adding new data that proved the roles of these proteins in replication, and in the expected steps of replication, is essential.

Reviewer #1:

Influenza RNA replication, distinct from mRNA synthesis, involves the synthesis of cRNA (RNA complementary to the genome) first, and then vRNAs (viral RNA, minus-strand). Both cRNA and vRNA have 5' triphosphate ends such that both are initiated de novo, not via cap-snatching as in transcription. This paper reports the identification of two cellular proteins that act to stimulate influenza RNA-dependent RNA polymerase (made up of PB1, PB2, and PA), specifically to increase the second step of replication, the synthesis of new vRNA from cRNA, as assayed in reactions on short templates in vitro. The products were confirmed to have appropriate sense and 5' triphosphate ends. The proteins with this activity were identified as pp32 and APRIL. Both had been found in other proteomic or functional screens. The proteins were confirmed to only stimulate vRNA and not cRNA synthesis. Both were shown to bind vRdRP (and only as a trimeric complex) and not the viral protein NP. KD led to small reductions in viral RNA levels in vivo.

This work is of interest as identifying strand-specific initiation factors for influenza replication. The in vitro stimulation assays, the MS identification, and the assays of the recombinant proteins seem adequate, notably the strand specificity. The binding to the trimer also seems convincing and is an interesting result. The paper provides a significant and useful addition to the field.

Concerns:

It would be helpful to know if these factors bind to RNA first and then vRdRP or the other way around. Can they bind and protect the 3' ends of the template?

The knockdowns show only modest effects on viral RNA levels and the CHX experiments are particularly unconvincing. Thus the importance of the factors in vivo is not certain. Better experiments in this area would be helpful.

Reviewer #2:

The influenza polymerase synthesizes viral genomic and messenger RNA in a temporally coordinated fashions. How vRNA, cRNA and mRNA production is regulated, and the co-factors that do this, is an area of significant interest. Sugiyama, et al. identify APRIL and pp32 as host proteins that regulate polymerase function. Their biochemistry nicely shows that APRIL/pp32 exclusively stimulate vRNA synthesis from a cRNA template in vitro. Nonetheless, the authors lack cell-based data showing that these proteins function in the same way as physiologically relevant regulators during viral infection.

1) The in vitro biochemistry is fairly convincing and shows that APRIL/pp32 stimulate vRNA synthesis from a model cRNA template. These finds are never validated in cells, raising multiple questions:

a) There are no data describing the effect of APRIL/pp32 knockdown on virus replication. Do these co-factors function during the full viral life cycle (e.g. single- or multi-cycle replication) and in the context of a real cRNP?

b) Knockdown of APRIL/pp32 reduces all viral RNA products during infection, as opposed to their in vitro data showing APRIL/pp32 only stimulate vRNA synthesis. Controls show that mRNA production is unchanged, but they could not test cRNA vs vRNA in the knockdown cells. As the major claim of their paper is that APRIL/pp32 stimulate cRNA synthesis, this discrepancy between their results in vitro and in cells during infection needs to be addressed experimentally. Perhaps investigating the role of APRIL/pp32 on purified cRNPs (following techniques from York et al., 2013), or when using a cRNA reporter, could help address these gaps.

c) Do APRIL/pp32 only affect vRNA synthesis in cells, or might they stimulate svRNAs? This is important to differentiate in cells as both vRNA and svRNA are produced from the same template by the same polymerase, and svRNAs have been implicated in shifting the polymerase towards synthesizing vRNA (just as the authors' APRIL/pp32 factors do). All of the gels for their in vitro assays are cropped, so it is not possible to know if svRNAs are produced in vitro.

2) Binding experiments in Figure 4 were all done in cells over-expressing transfected APRIL/pp32. Experiments demonstrating binding between the polymerase and endogenous APRIL/pp32 are needed to determine if these interactions are biologically relevant.

Reviewer #3:

Using cell-free influenza polymerase derived by micrococcal nuclease treated vRNPS purified from virions the authors are unable to detect unprimed vRNA synthesis using a 53-mer cRNA template, whereas ApG primed synthesis of vRNA is observed. They are able to purify a fraction of nuclear extract from uninfected HeLa cells that promotes unprimed vRNA synthesis, but not unprimed cRNA synthesis from a vRNA template. The active host proteins are identified by mass-spec as APRIL and pp32 and the effect is recapitulated with purified recombinant APRIL and pp32. These proteins are homologous, have independent and additive stimulatory activity and have previously been identified as influenza polymerase interacting proteins. They are shown to only bind to the trimeric polymerase, preferably in the 'free state' (i.e. not part of an RNP). Finally the authors show that siRNA knockdown of APRIL or pp32 or both in HeLa cells with subsequent infection by influenza virus leads to significantly reduced levels of all viral RNAs, vRNA, cRNA and mRNA, compared to the control infection without knockdown. The authors conclude that APRIL and pp32 promote viral replication by stimulating unprimed vRNA synthesis from cRNA.

This study raises a number of questions:

1) The source of polymerase the authors used is micrococcal nuclease treated vRNPs purified from virions. This is a poorly characterized, presumably heterogeneous sample. It is not clear, and should be stated, if any purification is done after stopping the nuclease digestion with 3 mM EGTA. Is the nuclease and EGTA still present in downstream experiments? One can imagine the preparation is highly heterogeneous containing free polymerase, polymerase bound to the conserved 3' and 5' ends (which are protected from nuclease), nucleoprotein bound to RNA fragments, free nucleoprotein, etc. All these things could influence the fate of added cRNA or vRNA template in an artifactual way (e.g. free NP sequestering template RNA, etc.). The authors should in the main text discuss what they think the preparation really contains.

2) Is this polymerase preparation capable of doing cap-dependent transcription e.g. upon adding capped RNA? If not, why not? If it can, then the effect of APRIL and pp32 on transcription could have been tested in vitro.

3) As the authors discuss in the Discussion, un-primed vRNA-synthesis was previously demonstrated in the absence of exogenously added primer or host factors by other authors using different preparation methods of polymerase. In contrast, Sugiyama et al. do not detect any vRNA-synthesis under their experimental conditions and in the absence of RNA-primer or host-proteins. However, as the authors point out, reaction-conditions like the reaction-time or particularly the enzyme-concentration, might affect the results and conclusions. To compare data from individual experiments and laboratories, it would be beneficial to determine (and not just discuss in the Discussion) the (estimated) active enzyme-concentration used in the experiments and/or vary parameters, in particular the reaction-time to perhaps overcome some detection limits.

4) Irrespective of the above, the authors come up with APRIL and pp32 as putative host factors that enhance vRNA synthesis, consistent with other recent work that identifies these proteins as polymerase associated host factors. The critical experiment is then the demonstration of this activity in infected cells. However the authors only show that siRNA knockdown of APRIL or pp32 or both in HeLa cells with subsequent infection by influenza virus leads to significantly reduced levels of all viral RNAs, vRNA, cRNA and mRNA. They argue that this is consistent with a specific role of APRIL and pp32 in being required for enhancing vRNA synthesis (which then has a knock on effect on other viral RNAs) but there is no direct proof that this is the mechanism in infected cells (except that they show there is no clear affect directly on primary transcription). For clarity and consistency the authors should also show growth curves for virus in knockdown and control cells and repeat these experiments in one other more relevant cell line (HeLA cells are very particular).

5) The authors propose that the mechanism for enhanced vRNA synthesis could be related to the different mode of initiation of vRNA synthesis, which has been reported to involve internal initiation and then realignment. Yet in an infected cell vRNA synthesis occurs in the context of a cRNP particle, whereas the authors argue that APRIL and pp32 preferably binds free polymerase. But free polymerase does not do RNA synthesis so how can APRIL and pp32 bound to free polymerase influence an initiation process that occurs buried in the internal cavity of the polymerase which is part of a cRNP? This paradox needs to be further discussed/explained by the authors.

---

## [Author Response]

Reviewer #1:*[…] It would be helpful to know if these factors bind to RNA first and then vRdRP or the other way around. Can they bind and protect the 3' ends of the template?*

To examine whether both IREF-2 proteins, pp32 and APRIL, bind the viral RNA template, we performed the electrophoresis mobility shift assay (EMSA). As shown in Figure 4—figure supplement 1, the interaction between each IREF-2 and viral RNA could not be observed. The description for this result has been added in the revised manuscript (subheading “Interaction between IREF-2 and viral RNA polymerases”).

The knockdowns show only modest effects on viral RNA levels and the CHX experiments are particularly unconvincing. Thus the importance of the factors in vivo is not certain. Better experiments in this area would be helpful.

Possible reasons for such “modest effects on viral RNA levels” by the knockdown observed in Figure 5 had been already explained in the Discussion section of the original manuscript and the revised manuscript. We cannot agree with the comment “the CHX experiments are particularly unconvincing”, because the result is statistically clear (Figure 5), demonstrating that the primary transcript transcribed only from “incoming vRNP” in the presence of CHX is not affected by IREF-2 knockdown, while the transcription level in the absence of CHX (i.e. the transcript sum of the primary transcription and multi-round transcription from newly replicated vRNA template) was significantly reduced by the knockdown. The effects of the knockdown on the transcripts level in the presence or absence of CHX condition were verified by student’s t test.

As “better experiments in this area”, growth curve experiments were also carried out, and significant defect in progeny virus production could be observed by knockdown of IREF-2. This result was represented as Figure 5 in the revised manuscript, and explained in paragraph three, subheading “Effect of IREF-2 on viral RNA synthesis in infected cells” in the revised manuscript.

Reviewer #2:*1) The in vitro biochemistry is fairly convincing and shows that APRIL/pp32 stimulate vRNA synthesis from a model cRNA template. These finds are never validated in cells, raising multiple questions:*

*a) There are no data describing the effect of APRIL/pp32 knockdown on virus replication. Do these co-factors function during the full viral life cycle (e.g. single- or multi-cycle replication) and in the context of a real cRNP?*

The results shown in Figure 5 clearly demonstrated that IREF-2s play a role in replication, but not in transcription process. As pointed out here, the effect of IREF-2 on the “multi-cycle replication” was examined by growth curve experiments, and significant defect in progeny virus production could be observed by knockdown of IREF-2. This result was newly added as Figure 5 in the revised manuscript, and explained in paragraph three, subheading “Effect of IREF-2 on viral RNA synthesis in infected cells”.

*b) Knockdown of APRIL/pp32 reduces all viral RNA products during infection, as opposed to their in vitro data showing APRIL/pp32 only stimulate vRNA synthesis. Controls show that mRNA production is unchanged, but they could not test cRNA vs vRNA in the knockdown cells. As the major claim of their paper is that APRIL/pp32 stimulate cRNA synthesis, this discrepancy between their results in vitro and in cells during infection needs to be addressed experimentally. Perhaps investigating the role of APRIL/pp32 on purified cRNPs (following techniques from York et al., 2013), or when using a cRNA reporter, could help address these gaps.*

According to the comment, the method for cRNP isolation suggested by this reviewer (York etal., 2013) was too difficult to establish in our hands during this revision period. Instead, we tried to distinguish the effect of IREF-2 on vRNA and cRNA syntheses in vivo using mini-replicon-based influenza virus reporter system as alternatively suggested by this reviewer. Here, vRNA and cRNA of the viral reporter genome (termed as “vNS-Luc” and “cNS-Luc”, respectively) were individually supplied using “pPol I-vNS-luciferase vector” and “pPol I-cNS-luciferase vector”, respectively (Figure 6 panel A). At 24 hours after transfection of expression vectors for viral polymerases, NP, and viral reporter genome RNAs into Control knockdown (KD) and IREF-2-double KD 293T cells, the accumulation levels of vRNA, cRNA, and mRNA from the viral reporter gene were quantitatively determined by RT-qPCR (Figure 6, panel B, also included in the manuscript as Figure 5—figure supplement 1).

Author response image 1.**DOI:**
http://dx.doi.org/10.7554/eLife.08939.013

The viral reporter RNA levels of 3 species in this experiment were significantly decreased by IREF-2 KD as observed in infected IREF-2 KD cells (Figure 5). In the case of vNS-luciferase as a source of the viral genome (Panel B, left panel), the vRNA level appears to be more decreased by IREF-2 KD (12% of control) compared with the reduction of the cRNA level (23% of control). Based on the assumption that IREF-2 stimulates preferentially vRNA synthesis rather than cRNA synthesis, it is expected that the vRNA level is to be more reduced by IREF-2 KD than the cRNA level in these experiments. Meanwhile, all viral RNA levels were shown decreased almost comparably (46%-48% reductions) in the case of cNS-luciferase used as a source of the viral genome (Figure 6, panel B, right panel). We consider that a convincing difference in the reduction level of between vRNA and cRNA could not be observed enough to distinguish the effects of IREF-2 KD on vRNA and cRNA syntheses in these reporter experiments. The stimulation effect of IREF-2 could be observed only in the case of unprimed vRNA synthesis in in vitro experiments as shown in Figure 3, while both vRNA and cRNA (and also mRNA) levels were decreased comparably by IREF-2 KD in in vivo experiments (Figure 5, and also Figure 6 shown above). A possibility for why vRNA and cRNA syntheses cannot be distinguished each other in infected cells had already been explained in the Results section (subsection “Effect of IREF-2 on viral RNA synthesis in infected cells”), as follows: vRNA serves as a template for cRNA synthesis, and vice versa. Therefore, if vRNA synthesis is impaired, cRNA synthesis (and also mRNA synthesis) is affected due to interdependency. This theoretically correct explanation has been already observed in a previous study (Maier et al.; Virology 370, 194-204, 2006). In their study, certain site-directed mutants of vRdRP, “T173A” or “F176A” substitution in PA subunit of vRdRP, were shown to have a significant defect only in cRNA template binding, but not vRNA template binding. Due to this selective defect in the template binding property, these vRdRP mutants exhibited significant loss of vRNA synthesis activity from the cRNA template in their cell-free (in vitro) reaction, but cRNA synthesis activity from the vRNA template of the vRdRP mutants was not impaired (see Figure 5 in Maier et al., 2006). In addition, significant reductions of all 3 kinds of viral RNA species (vRNA, cRNA, and viral mRNA) were observed at comparable levels in in vivo experiments using these vRdRP mutants (see Figure 7 in Maier et al., 2006). This previous observation by Maier et al., using vRdRP mutants appears to be similar to our results through depletion of host-derived factors by knockdown (in Figure 3 and Figure 5). The previous literatures quoted for above explanation (Maier et al., 2006) were also provided in the revised manuscript (paragraph two, subheading “Effect of IREF-2 on viral RNA synthesis in infected cells”) and listed in the References.

The reason for these observations is not clearly explained. At least for mini-replicon system, the system contains the background level of vRNA or cRNA synthesized from vectors by RNA polymerase I. To distinguish vRNA and cRNA synthesis in vivo, another approach would be needed.

*c) Do APRIL/pp32 only affect vRNA synthesis in cells, or might they stimulate svRNAs? This is important to differentiate in cells as both vRNA and svRNA are produced from the same template by the same polymerase, and svRNAs have been implicated in shifting the polymerase towards synthesizing vRNA (just as the authors' APRIL/pp32 factors do). All of the gels for their in vitro assays are cropped, so it is not possible to know if svRNAs are produced in vitro.*

Small leader vRNA (svRNA) is 22-27 nucleotide-long RNA from 5’-terminal region of vRNA and known to be important for the regulation of vRNA replication, as mentioned by this reviewer. This has already been taken into account and described in the Discussion section in the original manuscript and also in the revised manuscript. In addition, according to the comment, we tried to address the possibility that the production of svRNA is also regulated by IREF-2. As shown in the figure below (Figure 7), not only full-length RNA product (53 nt) but also short RNA products (~25 nt; termed as “Short products 2”) could be observed by adding recombinant pp32 to the cell-free reaction, suggesting the possibility that svRNA production is also stimulated by IREF-2.

Author response image 2.**DOI:**
http://dx.doi.org/10.7554/eLife.08939.014

2) Binding experiments in Figure 4 were all done in cells over-expressing transfected APRIL/pp32. Experiments demonstrating binding between the polymerase and endogenous APRIL/pp32 are needed to determine if these interactions are biologically relevant.

According to this comment, the interaction between viral polymerases (vRdRP) and endogenous IREF-2 proteins were examined. It is confirmed that both endogenous IREF-2 proteins (pp32 and APRIL) interact with the vRdRP trimeric complex. This result is represented as Figure 4—figure supplement 2, and explained in the revised manuscript (paragraph two, “Interaction between IREF-2 and viral RNA polymerases”).Reviewer #3:*1) The source of polymerase the authors used is micrococcal nuclease treated vRNPs purified from virions. This is a poorly characterized, presumably heterogeneous sample. It is not clear, and should be stated, if any purification is done after stopping the nuclease digestion with 3 mM EGTA. Is the nuclease and EGTA still present in downstream experiments? One can imagine the preparation is highly heterogeneous containing free polymerase, polymerase bound to the conserved 3' and 5' ends (which are protected from nuclease), nucleoprotein bound to RNA fragments, free nucleoprotein, etc. All these things could influence the fate of added cRNA or vRNA template in an artifactual way (e.g. free NP sequestering template RNA, etc.). The authors should in the main text discuss what they think the preparation really contains.*

This mnRNP preparation method has been established in over 20 years ago (Seong and Brownlee; Virology 186, 247-260, 1992), and well characterized by their studies and following studies. Fundamental properties of the mnRNP activity in the presence or absence of primer, and with vRNA or cRNA template characterized in the previous report (see Figure 7 in Seong and Brownlee, 1992) and our study (see lanes 1, 2, 6 and 7 in Figure 3) were completely the same.

At least, MNase and EGTA present in the mnRNP preparation seem not to exert an inhibitory influence on the viral RNA synthesis in cell-free system: in a previous report (Seong et al.; J. Biochem. 111, 496-499, 1992), 2 kinds of virion-derived enzyme sources, mnRNP (termed as “MN enzyme” in their report) and MNase/EGTA-free CsCl-RNP (termed as “CS enzyme”), were compared each other. There was no significant difference in the property of the vRdRP activity between these two enzyme sources (see Figure 1 in Seong et al., 1992). This strongly suggests that MNase and EGTA used for the mnRNP preparation have no influence on the enzymatic activity. In addition, the heterogeneity of the vRdRP in mnRNP has been taken account in the previous reports (Seong and Brownlee; Virology 186, 247-260, 1992). The vRdRP with residual 3’- and 5’- terminal parts of the viral genome RNA (vRNA) protected from MNase digestion, can be detected as 14-21 nt-long short RNA products in a previous report (Seong and Brownlee, 1992) and also in our hands. Therefore, such short by-products can be easily distinguished from 53 nt long objective products synthesized from the exogenously added viral model RNA templates (i.e. “v53” and “c53”) used in this study. Related to these issues described above, brief explanation about mnRNP was added in the Results section (subheading “Purification of IREF-2 from nuclear extracts of uninfected cells”), and the previous literatures quoted for above explanation were also referred there and listed in the References.

On the other hand, the effect of NP involved in the mnRNP preparation is relatively uncertain. However, the possibility mentioned in this comment that free NP sequesters the viral RNA template can be ruled out due to the fact that robust ApG-primed RNA products were synthesized from the exogenously-added viral template even in the presence of the same amount of NP present in the mnRNP preparation used for our experiments (see lanes 1 and 6 in Figure 3). The explanation about the influence of NP in the mnRNP preparation was added in the Discussion section (paragraph three). The relationship between IREF-2 (pp32 and APRIL) and NP is not addressed in this study. Therefore, the possibility that NP is requisite for the IREF-2 regulation in vRNA replication process should be taken into acount, and this seems to be addressed by the cell-free viral RNA synthesis system using NP-free enzyme source such as recombinant vRdRP. These points are discussed in the Discussion section (paragraph three) in the revised manuscript.

2) Is this polymerase preparation capable of doing cap-dependent transcription e.g. upon adding capped RNA? If not, why not? If it can, then the effect of APRIL and pp32 on transcription could have been tested in vitro.

The enzyme source, mnRNP, used in our study is fully capable of the cap-dependent transcription, and according to this comment, cell-free viral transcription assay was performed. The result is newly represented as Figure 3 and explained in the revised manuscript (paragraph three, subheading “Identification of pp32 and APRIL as IREF-2”). IREF-2 (pp32) gave no effect on the cap-dependent (cap-snatching) transcription in cell-free (in vitro) reaction (see Figure 3 lanes 3-5 in the revised manuscript). This observation is consistent with the result obtained in infected cells (in vivo) as shown in Figure 5, which demonstrated that primary transcription from incoming vRNP was not affected by IREF-2 knockdown.

3) As the authors discuss in the Discussion, un-primed vRNA-synthesis was previously demonstrated in the absence of exogenously added primer or host factors by other authors using different preparation methods of polymerase. In contrast, Sugiyama et al. do not detect any vRNA-synthesis under their experimental conditions and in the absence of RNA-primer or host-proteins. However, as the authors point out, reaction-conditions like the reaction-time or particularly the enzyme-concentration, might affect the results and conclusions. To compare data from individual experiments and laboratories, it would be beneficial to determine (and not just discuss in the Discussion) the (estimated) active enzyme-concentration used in the experiments and/or vary parameters, in particular the reaction-time to perhaps overcome some detection limits.

The enzyme source, mnRNP was established in earlier study and had been well-characterized (Seong and Brownlee; Virology 186, 247-260, 1992). It is known that mnRNP has very weak activity for unprimed cRNA synthesis from vRNA template, but no activity for unprimed vRNA syntheis from cRNA template (Seong and Brownlee, 1992, and Seong et al.; J. Biochem. 111, 496-499, 1992). Such properties are exactly the same as our mnRNP (see lanes 1, 2, 6 and 7 in Figure 3). As examined by a previous study (Seong and Brownlee, 1992), any parameters such as reaction-time, amount of mnRNP, pH, temparature, etc., in our cell-free reaction system were optimezed and have been already confirmed to be appropriate condition (Watanabe et al.; J. Virol. 70, 241-247, 1996, and Momose et al.; J. Virol. 75, 1899-1908, 2001).

This reviewer recommended to compare the enzyme sources in the previous reports using recombinant vRdRP and those in this study using mnRNP, but it is practically impossible due to the following reasons: among two previous studies using recombinant vRdRP, the amount of vRdRP used by Deng et al. (J. Virol. 80, 2337-2345, 2006) is unclear due to no information described (only description about the volume of vRdRP used in their reaction). On the other hand, 100 nM of vRdRP prepared using baculovirus expression was used by Zang et al. (J. Biol. Chem. 285, 41194-41201, 2010). However, it is very difficult to set up the reaction using 100 nM of mnRNP in our cell-free reaction due to the limitation of the stock concentration of mnRNP, which is prepared from a large quantity of virions (approximately 10-30 nM of vRdRP in our and other mnRNP stocks). Furthermore, an excess amount of mnRNP in our cell-free reaction system, if possible, is known to exhibit the inhibitory effect on the viral RNA synthesis even in the case of the ApG-primed RNA synthesis (Galarza et al.; J. Virol. 70 2360-2368, 1996). An exact reason for that is uncertain, and it might be plausible that an excess amount of NP accompanied with the addition of large amount of mnRNP in the reaction resulted in the inhibition of the reaction as observed in a previous study by Galarza et al.

In the previous report (Zang et al., 2010), unprimed RNA products were shown to be sythesized at the comparable level of ApG-primed RNA products (see Figure 1 in Zang. et al., 2010), even though the length of their unprimed RNA products were abnormal (3 nt-shorther than template size). In another previous report by Deng et al., (2006), the unprimed and ApG-primed RNA products were shown in different panels, so that it is difficult to compare each other. In our study on the other hand, unprimed RNA products were not detected at all, while robust ApG-primed RNA products were observed (compare lanes 6 and 7 in Figure 3 in this study).

Collectively, it seems difficult to compare results obtained by fundamentally different conditions. In addition, even for recombinant vRdRP with different enzymatic properties prepared by different laboratories, the difference among them is not completely addressed yet

4) Irrespective of the above, the authors come up with APRIL and pp32 as putative host factors that enhance vRNA synthesis, consistent with other recent work that identifies these proteins as polymerase associated host factors. The critical experiment is then the demonstration of this activity in infected cells. However the authors only show that siRNA knockdown of APRIL or pp32 or both in HeLa cells with subsequent infection by influenza virus leads to significantly reduced levels of all viral RNAs, vRNA, cRNA and mRNA. They argue that this is consistent with a specific role of APRIL and pp32 in being required for enhancing vRNA synthesis (which then has a knock on effect on other viral RNAs) but there is no direct proof that this is the mechanism in infected cells (except that they show there is no clear affect directly on primary transcription). For clarity and consistency the authors should also show growth curves for virus in knockdown and control cells and repeat these experiments in one other more relevant cell line (HeLA cells are very particular).

According to this comment, growth curve experiments using A549 cell line derived from human alveolar basal epithelium were carried out repeatedly (three times). As a result, significant defect in the progeny virus production and accumulation from IREF-2 knockdown A549 cells can be observed. The result was represented as Figure 5 in the revised manuscript, and explained and discussed in the Results section (paragraph three, subheading “Effect of IREF-2 on viral RNA synthesis in infected cells”).

5) The authors propose that the mechanism for enhanced vRNA synthesis could be related to the different mode of initiation of vRNA synthesis, which has been reported to involve internal initiation and then realignment. Yet in an infected cell vRNA synthesis occurs in the context of a cRNP particle, whereas the authors argue that APRIL and pp32 preferably binds free polymerase. But free polymerase does not do RNA synthesis so how can APRIL and pp32 bound to free polymerase influence an initiation process that occurs buried in the internal cavity of the polymerase which is part of a cRNP? This paradox needs to be further discussed/explained by the authors.

A complex of IREF-2 with vRdRP in infected cells appears to be template-free as shown in Figure 4, and ready to bind viral RNA template. cRNA would be recruited to the IREF-2-vRdRP complex (IREF-2-cRNP formation), followed by initiation and elongation of vRNA synthesis reaction. In our interpretation, this IREF-2 protein complexed with cRNP seems to be released from the cRNP complex during vRNA replication process. Therefore, we consider that the interaction between IREF-2-vRdRP and cRNA seems transient, and dissociated IREF-2 might interact with another RNA-free vRdRP again. Thus, the majority of IREF-2-vRdRP complex in infected cells appear to be free of RNA and NP/NP bound to RNA. The explanation according to this reviewer’s comment has been already described in the original manuscript and also in the revised version (Discussion).